# Decoding hierarchical control of sequential behavior in oscillatory EEG activity

Atsushi Kikumoto, Ulrich Mayr*

University of Oregon, Eugene, United States

**Abstract** Despite strong theoretical reasons for assuming that abstract representations organize complex action sequences in terms of subplans (chunks) and sequential positions, we lack methods to directly track such content-independent, hierarchical representations in humans. We applied time-resolved, multivariate decoding analysis to the pattern of rhythmic EEG activity that was registered while participants planned and executed individual elements from pre-learned, structured sequences. Across three experiments, the theta and alpha-band activity coded basic elements *and* abstract control representations, in particular, the ordinal position of basic elements, but also the identity and position of chunks. Further, a robust representation of higher level, chunk identity information was only found in individuals with above-median working memory capacity, potentially providing a neural-level explanation for working-memory differences in sequential performance. Our results suggest that by decoding oscillatory activity we can track how the cognitive system traverses through the states of a hierarchical control structure.

DOI: https://doi.org/10.7554/eLife.38550.001

## Introduction

Whether we dance or compose music, write computer code, or plan a speech: A limited number of basic elements—dance moves, notes, code segments, or arguments—need to be combined in a goal-appropriate order. Following *Lashley (1951)*, many theorists believe that such sequential skills are achieved through a set of abstract, content-independent control representations that code the position of basic elements within shorter subsequences (i.e. chunks) or the position of chunks within a hierarchically organized plan (*Dehaene et al., 2015*; *MacKay, 2012*; *Rosenbaum et al., 1983*).

Hierarchical control representations have been proposed as the hallmark of complex, flexible behavior in humans, including motor or action control (*Collard and Povel, 1982*; *Cooper and Shallice, 2006*; *Rosenbaum et al., 1983*; *Schneider and Logan, 2006*), language (*Fitch and Martins, 2014*), or problem solving (*Carpenter et al., 1990*; *Miller et al., 1986*). However, in principle, serial order can also be established through more parsimonious mechanisms based on learning associative links between between consecutive elements (often referred to as 'chaining'), without having to assume content-independent representations (*Botvinick and Plaut, 2004*; *Davachi and DuBrow, 2015*). Empirically, it is exactly the abstract nature of serial-order control representations that has made it difficult to provide direct evidence for their existence, to distinguish them from chaining-based representations, or to characterize their functional properties.

The current work tests the general hypothesis that flexibly generated and explicitly instructed, sequential behavior is achieved through hierarchical control representations and that these representations are reflected in the pattern of rhythmic EEG (electroencephalogram) activity. This hypothesis is based on two sets of observations.

First, hierarchical control representations need to coordinate a large array of sensory, motor, and higher level neural processes. A number of results suggest that such dynamic, large-scale neural

*For correspondence:
mayr@uoregon.edu

Competing interests: The authors declare that no competing interests exist.

communication can be achieved through frequency-specific, rhythmic activity that coordinates local and global cellular assemblies (*Buzsaki, 2006*; *Fries, 2005*; *Helfrich and Knight, 2016*). For example, the power of cortical oscillations in the theta band (4 to 7 *Hz*) increases as a function of serial-order processing demands (*Gevins et al., 1997*; *Jensen and Lisman, 2005*; *Jensen and Tesche, 2002*; *Roberts et al., 2013*; *Sauseng et al., 2009*). Specifically, *Hsieh et al. (2011)* reported that short-term memory of the order of serially presented visual objects was associated with increased frontal theta-band power. In contrast, the memory of the items themselves was related to changes in posterior alpha power (8 to 12 Hz, e.g. *Foster et al., 2016*). Combined, these results indicate that both the content and the serial order of a sequence are represented in oscillatory activity. Furthermore, these results provide some evidence that activity in the alpha-band may code for basic elements and in the theta-band for how elements are ordered.

Second, goal-relevant properties induce large-scale neural responses that are specific to the contents of representations (*Barak et al., 2010*; *Stokes et al., 2015*). These neural response profiles in turn can be decoded with time-resolved, multivariate pattern analysis (MVPA) from the pattern of spectral-temporal profile of EEG or MEG activity (*Foster et al., 2016*; *Fukuda et al., 2016*). While the majority of this research has focused on the representation of basic perceptual properties, there is also initial evidence that more abstract information, such as decision confidence (*King et al., 2016*) or semantic categories (*Chan et al., 2011*) can be decoded from the pattern of activity distributed over the scalp. We test here the hypothesis that even abstract, serial-order control representations can be decoded from the EEG signal independently of the basic elements.

By tracking representational dynamics while sequential action unfolds, we can also address open questions about the architecture of serial-order control. Traditional models of hierarchical control are informed by the characteristic response time (RT) pattern during sequence production, with long RTs at chunk transitions and bursts of fast, within-chunk responses (*Collard and Povel, 1982*; *Rosenbaum et al., 1983*). This pattern may indicate that higher level, chunk representations are needed only during chunk transitions, after which control is handed down to representations that code for the position and elements within chunks. Alternatively, higher level codes may need to remain active while lower level representations are being used—in order to ward off interference from competing chunks, or to maintain the ability to navigate within the overall control structure once within-chunk processing has completed. Currently, no neural-level data are available to distinguish between these theoretical options.

A related goal is to identify the source of individual differences in sequential performance within the serial-order control architecture. Initial evidence points to the importance of working memory (WM) capacity as one limiting factor during sequential performance, at least for longer sequences (*Bo and Seidler, 2009*). However, little is known about the exact nature of these constraints. One possibility is that WM limits the quality of representations in a uniform manner. This would imply that decodability of all representations, no matter which type (i.e. content or structural), or on which level, is reduced for individuals with low WM capacity. Arguably, however, the chunking of larger sequences into smaller subunits occurs precisely to protect representations from WM-related constraints (*Brady et al., 2009*; *Cowan, 1988*; *Oberauer, 2009*). Thus, based on this view, individual differences in sequential performance should be determined by the degree to which higher level representations can be maintained, while action selection is controlled by within-chunk representations. This would lead to the prediction that low WM capacity selectively affects decodability of higher level representations of chunk identity and/or position, but does not affect lower level, within-chunk representations.

In humans, there is already some evidence from studies using short-term memory tasks and fMRI neuroimaging for ordinal position codes (*Desrochers et al., 2015*; *Heusser et al., 2016*; *Kalm and Norris, 2014*; *Lehn et al., 2009*). However, as recently argued by *Kalm and Norris (2017a)*, it can be problematic to interpret position-specific information in most standard paradigms. Specifically, with the combination of short-term memory tasks and the low temporal resolution of fMRI, it is difficult to rule out serial-position confounds, such as sensory adaptation, temporal distance from a context change (i.e. beginning of a probe period), or differences in processing load (i.e. higher retrieval demands or uncertainty at the start of a sequence or chunk). Further, even if position codes can be validly decoded from fMRI signals, it is difficult to draw strong conclusions about the temporal dynamics between different serial-order control representations. Similarly, while fMRI neuroimaging studies have yielded important insights about the neural basis of different levels within a hierarchical

control structure (*Badre and D'Esposito, 2009*; *Farooqui et al., 2012*; *Koechlin and Summerfield, 2007*), we know very little about when in the course of sequential performance, which control representations are active. By using EEG signals, we are in a better position to track the temporal dynamics of serial-order control codes than when relying on fMRI analyses.

With our experimental paradigm, we tried to create a situation in which the use of content-independent control codes was particularly likely and that allowed us to minimize the serial-position confounds identified by *Kalm and Norris (2017a)*. In our procedure, participants were instructed to 'cycle through' a particular 9-element sequence (or six elements in Experiment 3) repeatedly within a block of trials (for a similar procedure using fMRI, see *Desrochers et al., 2015*). In each block, a new sequence was instructed, but all sequences consisted of recombinations of the same, small set of three basic elements (see *Figure 1*). Such a situation provides very little opportunity for an associative chaining process to control sequential performance. In fact, previous work had provided strong behavioral evidence that within this paradigm, serial-order control is achieved through hierarchically organized, content-independent position codes (*Mayr, 2009*).

## Results

### Overview

We measured EEG while participants performed an explicit sequencing task that was modeled after the task-span paradigm (*Mayr, 2009*; *Schneider and Logan, 2006*). *Figure 1a* shows an example of the sequences and the abstract sequence structure used in Experiments 1 and 2. The figure also illustrates the terminology for the different types of control representations we use throughout the paper

For each block of trials, participants were instructed to first memorize a new sequence of line orientations (see *Figure 1a* and Task and Stimuli in the Materials and methods section for details), and then 'cycle through' these memorized sequences on a trial-by-trial basis (i.e. starting over again when reaching the end of the sequence) for six (Experiments 1 and 2) or seven (Experiment 3) sequence repetitions per block. On each individual trial, participants had to respond per keypress whether or not the memorized orientation for that sequence position matched with a line orientation presented on the screen (see *Figure 1b*). As illustrated in *Figure 1a*, sequences had a two-level, hierarchical structure. Level 1 consisted of three ordered elements (i.e. 45°, 90°, or 135° orientations). To arrive at a complete sequence, on Level 2, we used either three (Experiments 1 and 2) or two (Experiment 3) ordered chunks. Thus, assuming the existence of Lashley-type control codes, we should expect on Level 1, independent representations that code for the content of the basic sequential elements (i.e. orientations) and for the position of each element within a chunk. On Level 2, we expect representations that code for the identity of each chunk and also for its position within the complete sequence. Across the sequences experienced by a given subject, the four control codes varied in a completely orthogonal manner. For example, each element occurred at each within-chunk position and between-chunk position equally often, just as each chunk identity occurred at each chunk position. This ensured that we could decode the serial-order control codes independently of each other.

As mentioned in the Introduction, identifying neural indicators of sequential representations, in particular position codes, is particularly challenging because there may be confounding variables that differ across positions within a sequence (*Kalm and Norris, 2017a*). The cycling-sequence paradigm eliminates some of the position confounds, such as sensory adaptation or the temporal distance from an encoding or retrieval phase. However, another set of confounds arises because early positions within a chunk are likely to invoke higher retrieval or WM demands. To counter such processing-demand confounds, we used in Experiment 2 a self-paced procedure, where participants were asked to move per key press on to the next position in the sequence, only *after* they retrieved the upcoming element. The rationale here is that the self-paced retrieval time should absorb processing demand effects and remove them from the actual preparation period (during which EGG registration occurred). In Experiment 3, we returned to a fixed-paced procedure, but reduced processing-demand effects by providing more robust pretraining of each chunk, prior to the actual test block. Also, across all experiments, participants were instructed (and encouraged through mild time pressure) to be ready for the next element prior to the probe. Therefore, we also separately

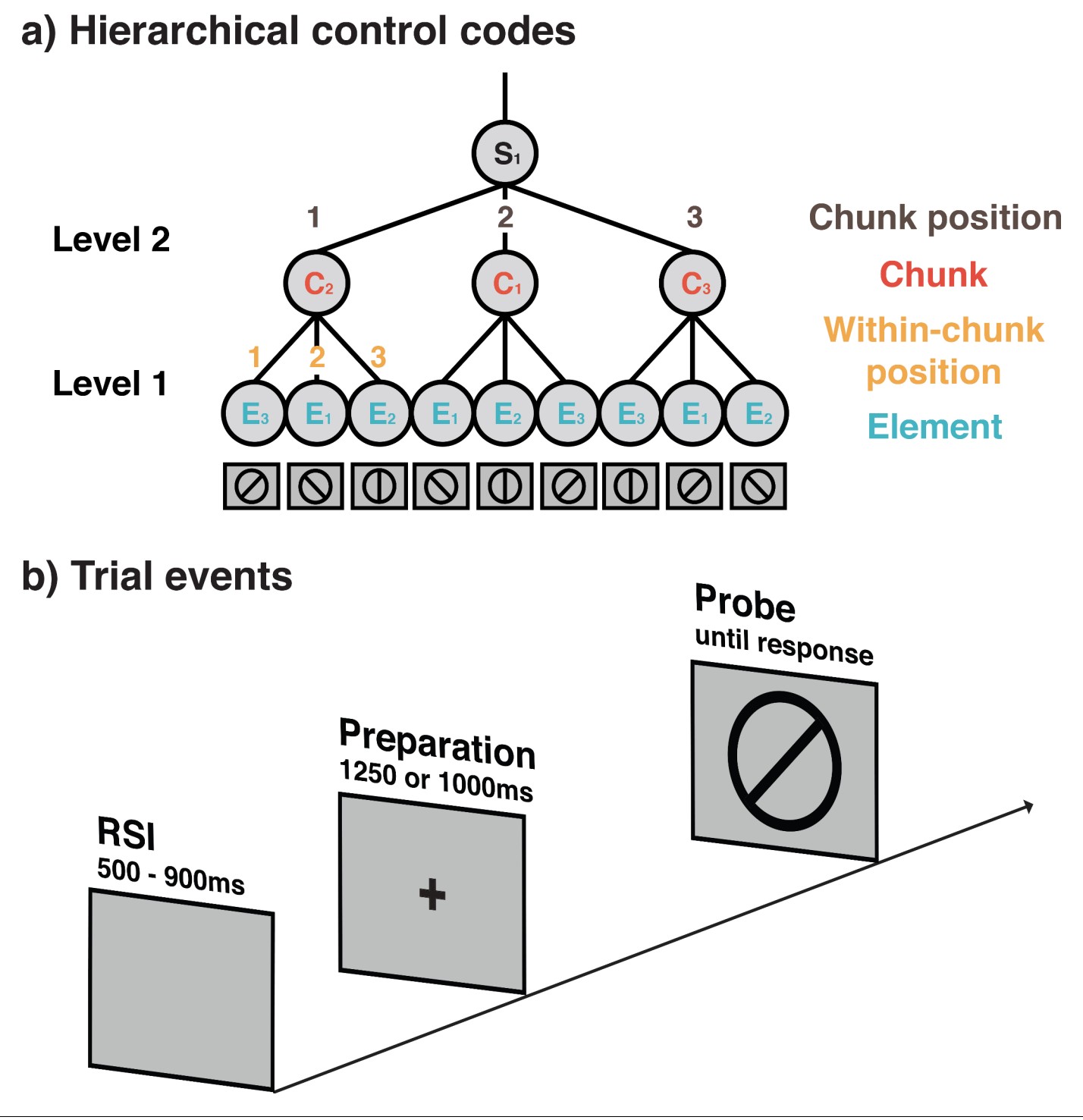

**Figure 1.** Schematic representation of hierarchically organized control codes for a sequence used in Experiments 1 and 2 (**a**) and the seqeunce of trial events (**b**). (**a**) *Example of the 9-element sequences used in Experiments 1 and 2 and the proposed control structure*. Sequences in these experiments were constructed from three different elements (orientations), three possible positions within chunks, three possible chunks, and three possible chunk positions. For each block, a new sequence following this general structure was instructed and had to memorized before the block was initiated. During instruction, memorization was aided by presenting each sequence spatially organized in terms of its different chunks. (**b**) *Structure of individual trials in Experiments 1 and 3*. Trials in Experiment 2 included a self-paced retrieval period prior to the preparation period. In all experiments, participants were asked to make keypress yes/no responses depending on whether or not the line orientation relevant for the current sequence position matched with

*Figure 1 continued on next page*

*Figure 1 continued*

the line orientation probe, presented on the screen. Importantly, on regular, non-error trials the screen provided no additional cues about the current position in the sequence.

DOI: https://doi.org/10.7554/eLife.38550.002

analyzed the preparation and the probe period, based on the assumption that even if the preparation period might be contaminated by processing-load confounds, these should be much less likely to occur during the probe period.

In Experiments 1 and 2, all sequences were constructed from just three different chunks. The use of three chunks allowed us to establish a level playing field for decoding all different types of codes with three elements, three within-chunk positions, three chunks, and three chunk positions, including equal representation of all possible combinations of these features across sequences. In particular, the use of three instances of each of the four control codes allowed us to have each of these instances present in every sequence. This is important from a decoding perspective as it removes temporal context (i.e. block) as a confounding factor in the decoding analyses. At the same time, the use of only three chunks and the constraint of counterbalancing individual elements (i.e. orientations) across positions, resulted in within-chunk sequences that contained the same element-to-element transitions across chunks (albeit at different within-chunk positions; e.g. *ABC* and *CAB*). In principle, the representation of such sequences could be supported through a simple chaining mechanism that requires no abstract position codes. In order to generalize our position decoding results to a situation that did not allow chaining, we used in Experiment 3 six possible chunks. Here, participants experienced each possible element-to-element transition (excluding element repetitions) equally often (i.e. ABC, ACB, BAC, BCA, CAB, ABA), rendering a simple chaining mechanism that picks up on across-sequence regularities useless.

In order to identify the representations that control sequential performance, we performed a series of decoding analyses with the spectral-temporal profiles of EEG activity over the scalp. We examined the entire preparation period prior to the appearance of the test probe (i.e. 1250 ms for Experiment 1, 1100 ms for Experiment 2) and the initial 300 ms following the test probe. While we present decoding results across the entire preparation and probe interval, for the preparation interval we focus mainly on the 600 ms prior to the probe onset in order to reduce potential effects of condition-specific retrieval confounds. At each time sample during these periods, the pattern of rhythmic activity at a specific frequency value (from 4 Hz to 35 Hz) was used separately to train linear classifiers via a four-fold cross-validation procedure (see Frequency-by-Time Decoding Analysis in Materials and methods section for details). This generated a matrix of frequency- and time-resolved decoding results for each of the task-relevant representations: (1) elements, (2) within-chunk ordinal positions, (3) chunk identities, and (4) chunk positions (*Figure 1a*).

We had no a-priori predictions about which electrodes would contribute to the decoding results and therefore we report no detailed information about the topography of decoding results (for one exception, see *Within-Chunk Positions and Processing-Demand Confounds* in the Results section). However, for sake of completeness, we provide information about how different electrodes contribute to decoding accuracy in the Appendix.

## Experiments 1 and 2

Experiments 1 and 2 used very similar procedures and therefore will be reported together. RTs from error-trials, post-error trials, and trials in which RTs were shorter than 100 ms were excluded from analyses. For Experiment 2, trials with a self-paced retrieval time longer than 8000 ms were excluded. To analyze the effects of ordinal positions on serial-control performance, we specified two sets of orthogonal contrasts for positions within chunks, the first comparing position 1 against positions 2 and 3, the second comparing positions 2 and 3.

## Behavior

We examined the pattern of RTs and errors in order to confirm that participants relied on a hierarchical control structure to perform the sequencing task (*Figure 2*). Specifically, we examined to what degree participants were slower and less accurate at the first position of a chunk, compared to the

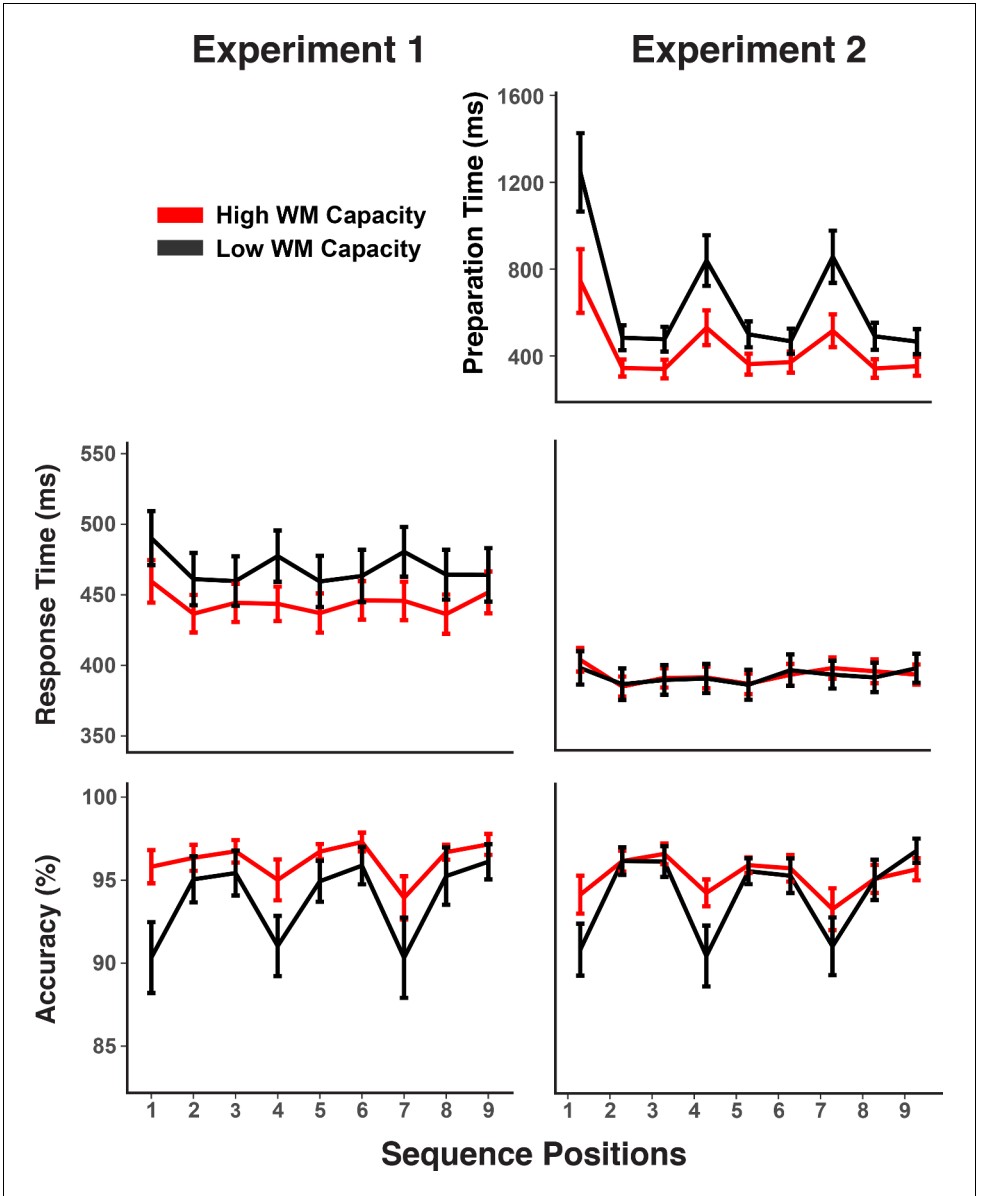

**Figure 2.** Experiments 1 and 2 behavioral results. RTs and accuracy show marked sequence structure effects in Experiment 1. In Experiment 2, a substantial portion of the RT sequence structure effect is shifted into the self-paced retrieval period. Error bars show within-subject 95% confidence intervals.

DOI: https://doi.org/10.7554/eLife.38550.003

remaining positions. Note, that for both experiments, but in particular for Experiment 2 we had emphasized to participants the need to prepare for each upcoming probe. Thus, compared to previous work with similar sequencing tasks (e.g. *Mayr, 2009*), we expected rather subtle structure effects in RTs, whereas we expected errors to remain sensitive to the greater difficulty of activating the correct chunk during chunk boundaries. For Experiment 1, which used the fixed-paced procedure, RTs showed a small, but reliable chunk boundary effect of 14 ms ($SD$ = 18) in RTs, $F$(1,28) = 19.20, $MSE$ = 622.47, p < 0.001, and a robust error effect of 3.8% ($SD$ = 4.3), $F$(1,28) = 22.70, $MSE$ < 0.01, p < 0.001. For Experiment 2, where participants moved to the next trial only after indicating per key-press that they had retrieved the upcoming orientation, the chunking pattern in RTs, while still reliable was with 5 ms ($SD$ = 9.3) was even smaller than in Experiment 1 for RTs, $F$(1,28) = 8.05, $MSE$ = 168.79, p = 0.01, but with 3.5% ($SD$ = 4.9%) equally strong for errors, $F$

(1,28) = 18.41, *MSE* < 0.01, p < 0.001. There was a very strong chunk-boundary effect of 372 ms (*SD* = 361) in the self-paced retrieval times, $F_{(1,28)}$ = 35.43, *MSE* = 234826, p < 0.001. Thus, overall the self-paced retrieval instruction was effective in removing an even larger share of the retrieval demands from the EEG-recorded preparation period. At the same time, the additional opportunity for self-paced retrieval did not change the fact that participants made more retrieval errors at transition points.

*Figure 2* also indicates that individuals with low WM capacity (i.e. determined through median split) showed larger chunk-boundary effects than individuals with high WM capacity. To increase statistical power for detecting such individual-differences effects within our design, we combined accuracy scores across both experiments. The WM-group by chunk-boundary interaction was reliable across both experiments, $F_{(1,56)}$ = 6.63, *MSE* < 0.01, p = 0.01, and it was close to reliable in each of the individual experiments, Exp. 1: $F_{(1,28)}$ = 3.24, *MSE* < 0.01, p = 0.08, Exp. 2: $F_{(1,28)}$ = 3.40, *MSE* < 0.01, p = 0.08. Given that the use of self-paced retrieval in Experiment two eliminated much of the retrieval demands from RTs, a meaningful test for RT effects can be conducted only in Experiment 1, where the WM-group by chunk-boundary interaction was in the predicted direction, albeit not reliable, $F_{(1,28)}$ = 1.51, *MSE* < 168.79, p = 0.22. For Experiment 2, the WM-group by chunk-boundary interaction was nearly reliable for self-paced retrieval times, $F_{(1,28)}$ = 4.19, *MSE* = 976921, p = 0.05. Thus, across experiments there is a consistent pattern of people with lower WM capacity showing greater difficulty at chunk boundaries.

## Basic elements: orientations

Three different line orientations served as the basic elements of the pre-instructed sequences that had to be retrieved and actively maintained in WM for the comparison with the test probe. Based on previous work, we hypothesized that the spatial pattern of alpha-band (8–12 Hz) oscillations across electrodes contains information about orientations. Therefore, we decoded the three instances in which basic elements could occur (i.e. 45° vs. 90° vs. 135° orientation) over the range of frequency values and time samples (see *Frequency-by-Time Decoding Analysis* in *Results* section for details). Indeed, we found across Experiments 1 and 2 that oscillations centered around the alpha-frequency band showed sustained decoding of orientations, both late in the preparation period and early in the probe period (cluster-forming threshold p < 0.05, corrected significance level p < 0.001; *Figure 3a*). These results are consistent with the previous reports that specific WM representations can be decoded from scalp-distributed alpha activity while maintain in working memory (*Foster et al., 2016*; *Fukuda et al., 2016*). A novel aspect of the current results is that we found alpha-encoded representations of stimuli that were retrieved from long-term memory rather than maintained in working memory. In addition, during the early probe period, the current element was also decoded from theta-band (4–7 Hz) activity. This result is consistent with previous findings in which stimulus-evoked theta oscillations encoded spatial properties of visually presented stimulus (*Foster et al., 2016*).

These decoding results do not provide causal information about the relevance of the proposed control codes for behavior. However, we can check to what degree element decoding accuracy predicts trial-by-trial variability in performance over and above other potentially relevant control codes (e.g. position codes; see Multi-level Modeling in the Materials and methods section for details). We first residualized all RTs with regard to potential nuisance variables. Then, we submitted the residualized RTs to a multi-level regression model that included trial-by-trial indicators of decoding accuracy for element codes in the alpha band and position codes in the theta band as predictors (see next section). During the probe period, decoding accuracy for elements in the alpha band predicted faster trial-to-trial responses over and above other constructs (i.e., within-chunk positions and chunk identity), *b* = –032, *SE* = 0.09, *t* = –3.28, for Experiment 1, and, *b* = –016, *SE* = 0.04, *t* = –3.51, for Experiment 2. There were no significant predictive effects during the preparation period for elements, *b*= –0.03, *SE* = 0.014, *t* = –0.27, for Experiment 1, and, *b* = 0.07, *SE* = 0.001, *t* = 0.73, for Experiment 2.

## Decoding within-chunk positions

One of our central questions is to what degree we can find evidence for content-independent representations of the current within-chunk position. Based on the previous literature, we hypothesized

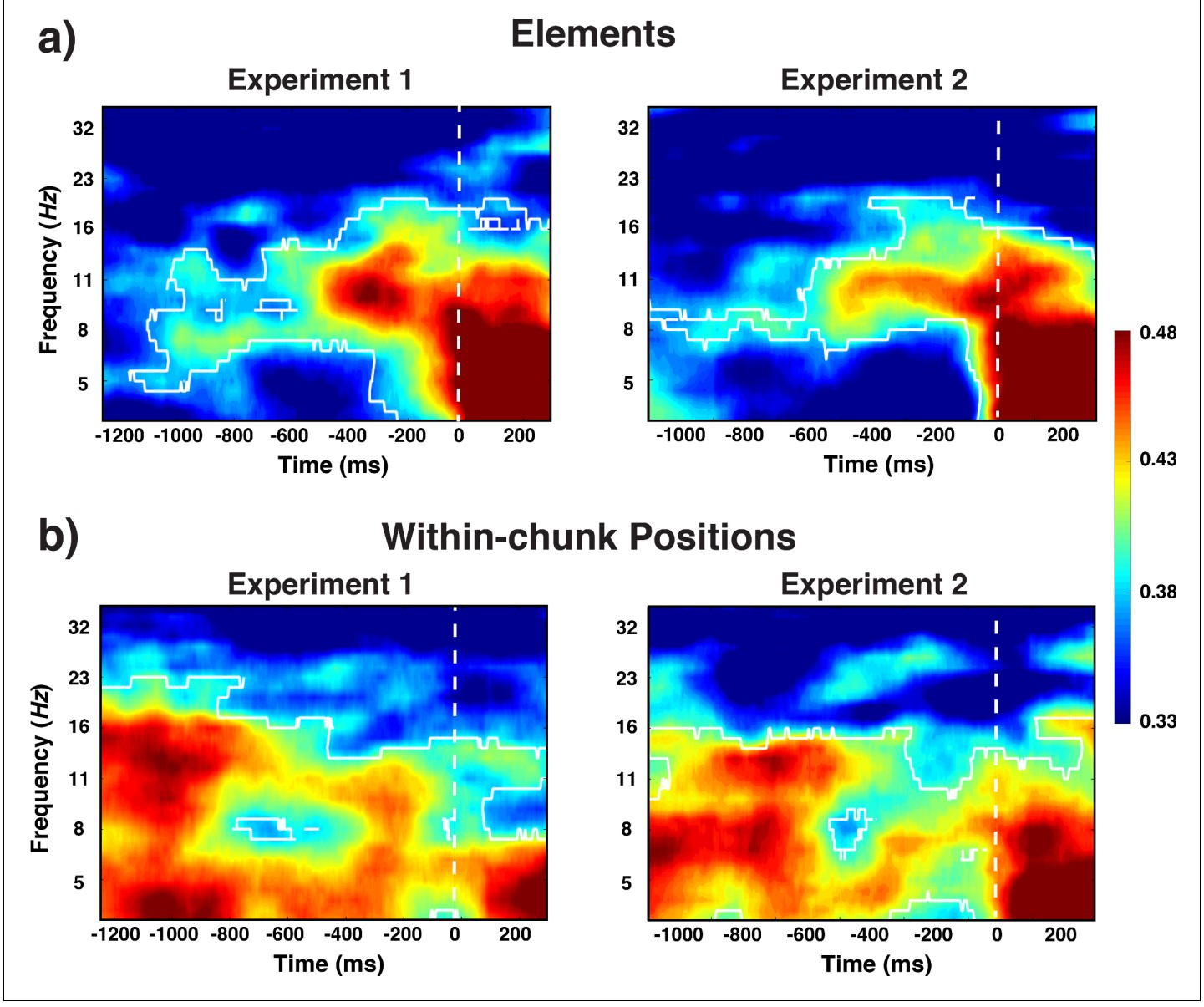

**Figure 3.** Experiment 1 and 2 decoding results as a function of frequency and time relative to probe-stimulus onset for elements/orientations. (a) and within-chunk positions (b). The legend shows the decoding accuracy in probability. Chance level is p = 0.33. The regions enclosed by white lines show significant decoding accuracy after cluster correction.

DOI: https://doi.org/10.7554/eLife.38550.004

that theta-band (4–7 Hz) oscillations are a prime candidate as a carrier of such information (*Hsieh et al., 2011*). We decoded here the three instances of within-chunk positions (first vs. second vs. third position in a chunk). Across Experiments 1 and 2, our decoding results showed that within-chunk position information was present in a broader band of oscillations (4–16 Hz), but with a particular emphasis on the theta-band (cluster-forming threshold p < 0.05, corrected significance level p < 0.001; *Figure 3b*). In the theta-band, robust decoding accuracy was observed at the beginning of the preparation and probe period, whereas decoding accuracy gradually decreased in the alpha-band as the probe period approached.

The same multi-level model used to test the predictive effects of element representations for RTs (see previous section) had also included a predictor reflecting position-decoding accuracy. For Experiment 1, more robust evidence for position information in the theta-band during the probe

period predicted faster RTs, $b = -0.011$, $SE = 0.0047$, $t = -2.44$. The theta-band effect was in the same direction, but not reliable in Experiment 2, $b = -0.005$, $SE = 0.0044$, $t = -1.12$. It is possible that the overall reduced variability of RTs due to the self-paced procedure (see *Figure 2*) may have rendered this experiment less sensitive to trial-to-trial variability in performance than Experiment 1.

## Generalizing within-chunk positions

A key feature of Lashley-type position codes is that they are independent of other control codes, such as those coding for the basic elements the sequences (i.e. orientations). One way to establish the independence of decoded information is to demonstrate that it generalizes across theoretically critical context variation. For example, to examine the independence of position codes from element codes, we trained the decoders to classify positions for two of three possible elements (e.g. 45° and 90° orientations) and then tested generalization of decoders for the remaining element (e.g. 135° orientation), repeating this process until each element was tested for generalization (see Generalization Analysis in the Materials and methods section for details). As a reference for the generalization scores we also applied the same decoders to test sets consisting of the same instances as the training sets (e.g. 45° or 90° orientation) that had been excluded from the training set. In the same manner, we also conducted generalization analyses of position codes across different instances of chunk positions.

*Table 1* shows results from these analyses for Experiments 1 and 2, separately for generalization across elements and chunk positions, and for both the preparation phase (i.e. averaging across a 600-ms interval prior to stimulus presentation) and the probe phase (averaging across a 300 ms interval post stimulus presentation). In each case, generalization accuracy was essentially indistinguishable from reference scores, indicating that position codes were indeed fairly independent from either specific elements or chunk positions.

Note, that in Experiments 1 and 2 we were not able to examine generalization across different chunk identities, as within-chunk-position/element combinations were not independent of chunk

**Table 1.** Average generalization scores (SE) for the decoding of within-chunk position codes.

| | Preparation | | | |
| --- | --- | --- | --- | --- |
| | **Theta** | | **Alpha** | |
| Generalization variable | Reference | Generalized | Reference | Generalized |
| Experiment 1 | | | | |
| Elements | 0.381 (0.003) | 0.377 (0.003) | 0.378 (0.004) | 0.370 (0.004) |
| Chunk positions | 0.384 (0.004) | 0.387 (0.004) | 0.374 (0.002) | 0.375 (0.002) |
| Experiment 2 | | | | |
| Elements | 0.367 (0.004) | 0.363 (0.004) | 0.369 (0.005) | 0.360 (0.005) |
| Chunk positions | 0.369 (0.003) | 0.373 (0.003) | 0.369 (0.004) | 0.362 (0.004) |
| | Probe | | | |
| | Theta | | Alpha | |
| Generalization variable | Reference | Generalized | Reference | Generalized |
| Experiment 1 | | | | |
| Elements | 0.394 (0.005) | 0.385 (0.005) | 0.362 (0.007) | 0.364 (0.007) |
| Chunk positions | 0.388 (0.005) | 0.383 (0.005) | 0.361 (0.004) | 0.358 (0.004) |
| Experiment 2 | | | | |
| Elements | 0.404 (0.003) | 0.394 (0.005) | 0.378 (0.005) | 0.378 (0.005) |
| Chunk positions | 0.397 (0.006) | 0.383 (0.006) | 0.375 (0.005) | 0.376 (0.005) |

*Note.* The table shows for each generalization variable, position-code generalization scores and corresponding reference scores (see Generalization Analysis in the Materials and methods section for details). Results were averaged for either the preparation or the probe period across time (i.e. 600–0 ms for the preparatory interval, 0–300 ms for the probe interval) and frequency values within theta (4–7 *Hz*) and alpha (8–12 *Hz*) bands.

DOI: https://doi.org/10.7554/eLife.38550.005

identity. However, we revisited this issue in Experiment 3, where there was no confound between chunks and position/element combinations.

## Within-chunk positions and processing-demand confounds

One important question is to what degree processing-demand confounds across within-chunk positions (e.g. retrieval demands) may produce position-like decoding results. Retrieval demands and supposedly also other processing demands should be considerably reduced in the self-paced procedure used in Experiment 2, relative to the fixed-paced procedure in Experiment 1. Thus, the fact that the decoding results were very similar across the two experiments speaks against the role of such confounds. Nevertheless, to further probe the possible contribution of position-related demand effects, we conducted two additional sets of analyses.

First, we used the fact that mid-frontal theta power has been shown to reflect cognitive control demands and therefore should be sensitive to a position/control demand confound (*Cavanagh and Frank, 2014*; *Cohen and Donner, 2013*). Thus, in principle, larger control demands during position one than the remaining within-chunk positions could lead to corresponding variations in theta power, and thereby contribute to the decoding of positions. Conversely, a finding of no cross-position differences in mid-frontal theta power would rule out one plausible way in which a load confound might affect decoding results. Comparisons of average theta power at mid-frontal electrodes (Fz and Cz electrodes) between the first and the remaining two, within-chunk positions revealed no reliable differences for either the preparatory or the probe period, $t = 1.00$, $b = 0.49$, $SE = 0.49$, and $t = 0.17$, $b = 0.12$, $SE = 0.70$, for Experiment 1; $t = 0.59$, $b = 0.17$, $SE = 0.29$ and $t = -0.16$, $b = -0.07$, $SE = 0.45$ for Experiment 2. We also looked at other electrodes and found no reliable differences. These results indicate that position decoding in the theta band does not simply reflects univariate and consistent changes across individuals that are solely driven by the modulation of frontal theta power. They also indicate that the observed, position-decoding results are unlikely to be driven by a confound between positions and control/retrieval demands.

Second, we examined the implications of a processing-load and a position-decoding explanation on our decoding results using representational similarity analysis (RSA, *Cichy et al., 2014*; *Grootswagers et al., 2017*; *Kriegeskorte et al., 2008*). RSA tests the similarity structure of neural responses that underlie the decoding results in a model-driven manner (see Representational Similarity Analysis in the Materials and methods section). For this purpose, we examined the confusion matrices across decoded positions. Arguably, if the decoding results reflect discrete position codes, one might expect that the pattern of EEG activity is distinct across all three positions (i.e. a uniform pattern of confusions across the off-diagonal cells; see *Figure 4a*). In contrast, if heightened retrieval/processing demands during position one drive the decoding results then position one might be particularly distinct from positions 2 and 3, whereas the latter two positions should be more similar to each other (see *Figure 4a*). As shown in *Figure 4b*, we found that for the preparation period in both Experiments 1 and 2, position one was particularly distinct, albeit with some remaining, numerical differences between positions 2 and 3. However, during the probe period, the three positions appeared to be equally dissimilar to each other. As a statistical test of these observations, we used hierarchical, linear regression models to predict classification probabilities simultaneously with both the unique-position-1 model matrix and the discrete-position model matrix as predictors (see *Figure 4a*). During the preparation period, only the unique-position-1 model significantly predicted the observed confusion matrix (unique-position-1: $b = 1.48$, $SE = 0.28$, $t = 5.36$; discrete: $b = -.10$, $SE = 0.16$, $t = -.59$, for Experiment 1, unique-position-1: $b = 1.10$, $SE = 0.21$, $t = 5.16$; discrete: $b = 0.08$, $SE = 0.13$, $t = 0.67$, for Experiment 2). However, only the discrete model significantly predicted the observed pattern during the probe period (unique-position-1: $b = 0.21$ $SE = 0.29$, $t = 0.73$; discrete: $b = 0.75$, $SE = 0.23$, $t = 3.20$, for Experiment 1; unique-position-1: $b = 0.16$, $SE = 0.36$, $t = 0.44$; discrete: $b = 1.01$, $SE = 0.30$, $t = 3.40$, for Experiment 2).

For the preparation period, the pattern of decoding results was ambiguous with regard to the question whether they truly reflect characteristics of the sequential representation or position-correlated processing-demands. Importantly, a retrieval confound is not the only possible explanation for a unique-position-1 pattern. People may use less precise position representations during preparation (i.e. differentiating mainly between the beginning and later portions of a chunk) and arrive at a more fine-grained representation only during the probe period, when the actual response is imminent. The fact that we obtained similar results across Experiments 1 and 2, despite the very different

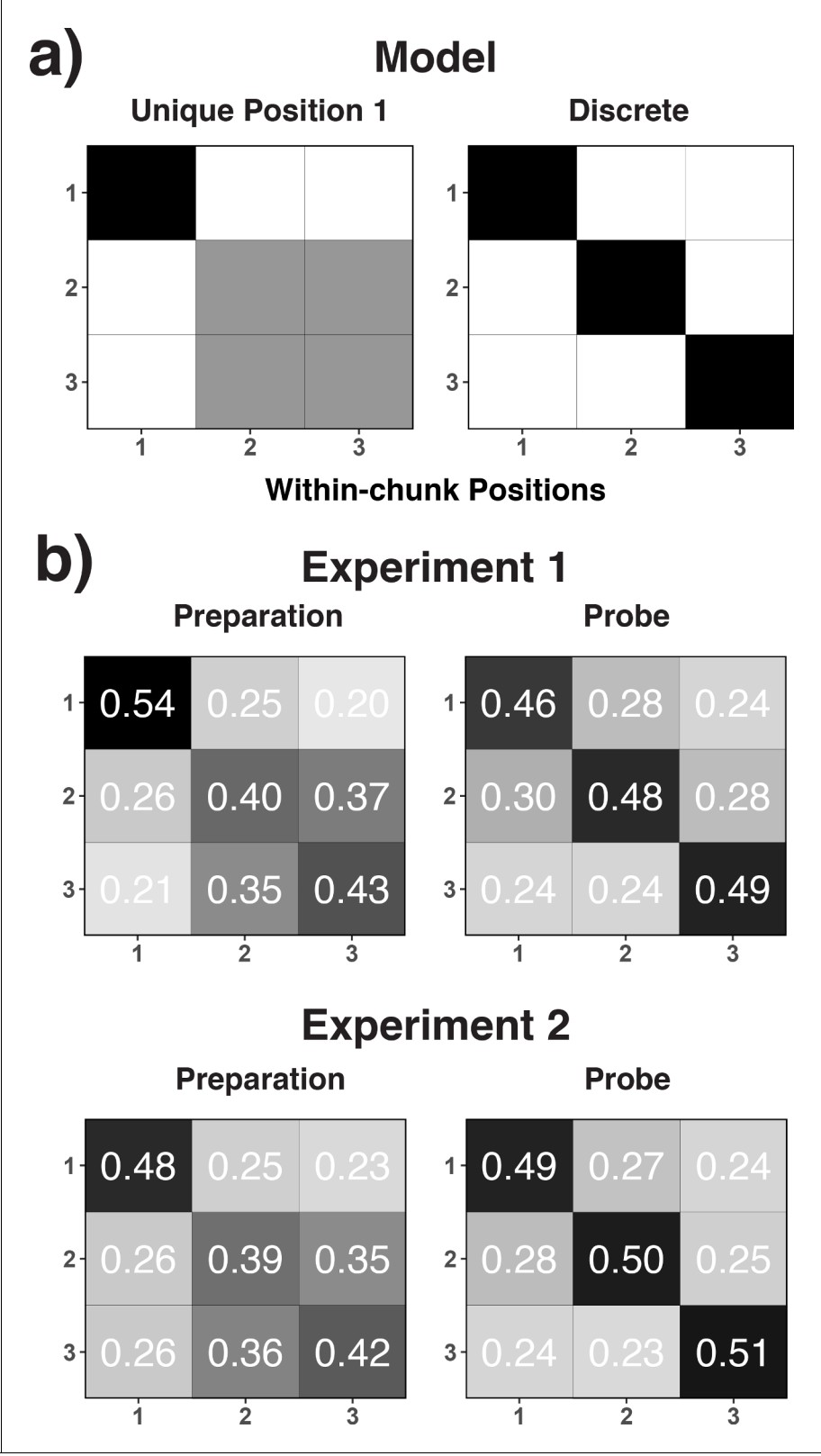

**Figure 4.** Theoretical (**a**) and empirical (**b**) confusion matrices to test the unique-position-1 model against the discrete-position coding model. (**a**) RSA model matrices used to test the unique-position-1 account against the discrete-position-coding account. In all presented confusion matrices, the x-axis represents the correct code and the y-axis represents the relative frequency with which each possible code is selected for that specific correct code (i.e. columns add up to 1.0). Black cells in the model matrices represent a theoretically expected correct classification probability of p = 1.0, gray cells of

*Figure 4 continued on next page*

*Figure 4 continued*

p = 0.5, and white cells of p = 0. (**b**) Confusion matrices for position decoding results for Experiments 1 and 2 are shown separately for the preparation and the probe period.

DOI: https://doi.org/10.7554/eLife.38550.006

processing demands during the preparation periods across these experiments, makes it more likely that these are true representational effects. Also, the fact that frontal theta power was modulated by within-chunk position neither during the preparation period, nor during the probe period speaks against the effect of processing demand confounds. Finally, additional evidence that is inconsistent with a processing-demand interpretation of the position-decoding effects will be provided when we analyze effects of working-memory differences on decoding results.

Irrespective of the interpretation of results during the preparation period, the probe-period results suggest that theta-activity represents actual position codes. The clear evidence of position coding during the probe period, but not the preparation period, is consistent with recent behavioral results indicating that people tend to update their current position within a larger sequence when they actually execute an individual, sequential element, not when they prepare for it (*Mayr et al., 2014*).

## Evidence for associative chaining?

The sequences in Experiments 1 and 2, used unique element-to-element transitions within chunks. In principle, this would allow representations based on associative chaining to emerge, and control sequential behavior. *Schapiro et al. (2012)* suggested as a test for associative chaining between elements *A* and *B* to compare the correlation between the neural pattern associated with *A* early in training and the pattern associated with *B* late in training with the correlation between A late in training and B early in training. The logic behind this test is that through experience with specific element-to-element associations, the representation of the current element drifts toward the representation of the upcoming element in an anticipatory manner so that $cor(A_{late}, B_{early})$ becomes larger than $cor(A_{early}, B_{late})$. For both Experiments 1 and 2, we computed these indicators of associative chaining separately for specific, within-chunk, element-to-element transitions and then averaged the relevant z-scored correlations across transitions and within individuals. In no case did we find any indication of a forward predictability pattern during the time- and frequency-ranges that contained robust element information (preparation period/alpha band: $t(30) = 0.739$, $p = 0.465$, probe period/alpha band: $t(30) = -1.60$, $p = 0.120$, probe period/theta band: $t(30) = -1.26$, $p = 0.218$, for Experiment 1; preparation period/alpha band: $t(30) = 1.07$, $p = 0.293$, probe period/alpha band: $t(30) = 0.912$, $p = 0.369$, probe period/theta band: $t(30) = 0.425$, $p = 0.674$, for Experiment 2). Strong inferences from these results rest on accepting the null hypothesis. Nevertheless, they are consistent with the conclusion that chaining was not a major factor in the current paradigm. Experiment 3 provided an opportunity for additional tests of predictions from associative chaining models.

## Decoding of chunk identity and position

In addition to the element-level representations, we examined to what degree information about Level-2 representations can be extracted from the EEG signal. In Experiment 1 and 2, three different chunks occurred at three different positions within sequences, across experimental blocks. Therefore, we tried to independently decode the three instances of chunk identity (ABC vs. BCA vs. CAB) and the three instances of chunk positions (first vs. second vs. third chunk in a sequence).

We found that information about the identity of chunks was decodable during the preparation period (cluster-forming threshold $p < 0.05$, corrected significance level $p < 0.05$; *Figure 5*). Across both Experiments 1 and 2, the effect was most sustained 600 ms prior to the onset of the test probe in the alpha-band (8–12 Hz). In Experiment 2, there was also above-chance decoding in the theta band around the time of stimulus representation; an effect that we will not further interpret given that it was not found in Experiment 1. Generally, only very little chunk-identity information was detectable during the probe period.

One important theoretical question is to what degree chunk representations are active mainly during chunk transitions (i.e. during the first position in each chunk) or are required equally across all

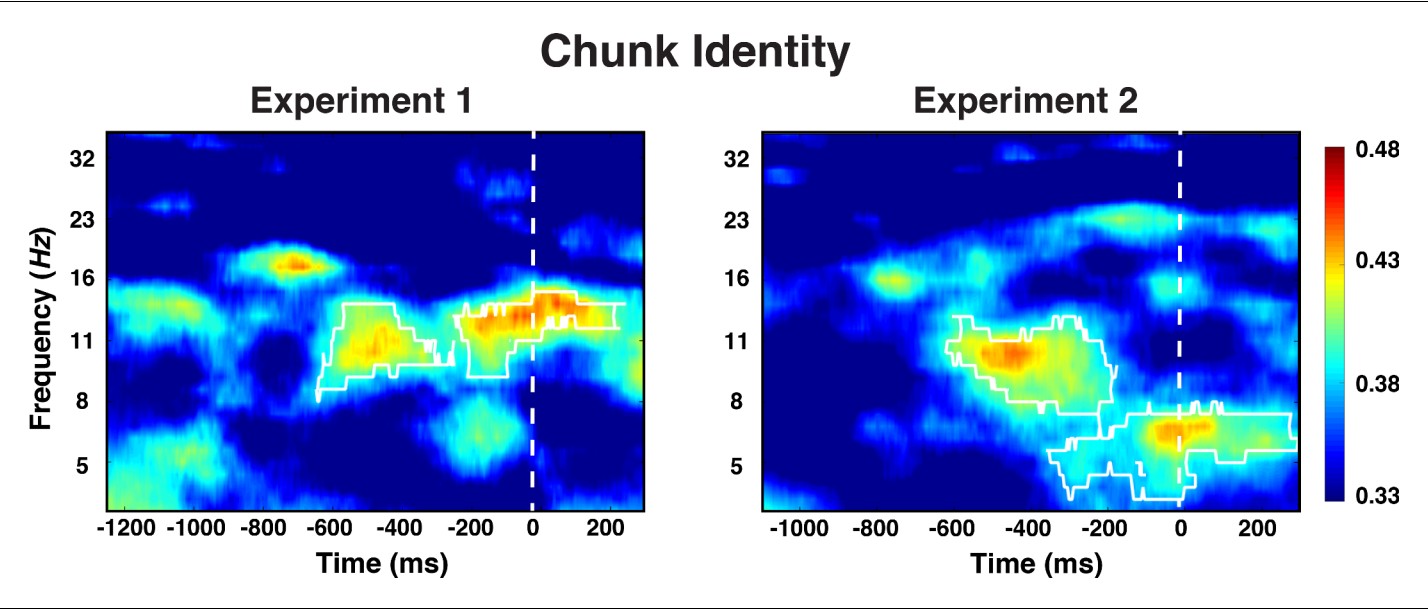

**Figure 5.** Experiment 1 and 2 decoding results as a function of frequency and time for chunk identity. The legend shows the decoding accuracy in probability. Chance level is p = 0.33. The regions enclosed by white lines show significant decoding accuracy after cluster correction.
DOI: https://doi.org/10.7554/eLife.38550.007

positions, potentially to constrain lower-level representations. To address this question, we examined to what degree the chunk-specific neural patterns recur across all three within-chunk positions. We summarized the decoding results for each position separately with classifiers trained on the data with all positions. As evident in the *Figure 6*, while there is some variability in the strength of expression of chunk information for positions 2 and 3 across the two experiments, there is no evidence that chunk information is present only for position 1. A contrast between positions 1 versus positions 2 and 3 produced a significant effect in neither experiment, Experiment 1: $t(30)$ = .02, p = 0.99, Experiment 2: $t(30)$ = .88, p = 0.38. The contrast between positions 2 and 3 did produce a reliable difference in Experiment 1, $t(30)$ = 2.17, p < 0.05, but not in Experiment 2, $t(30)$ = -1.12, p = 0.27. Thus overall, information about the current chunk is represented not only when a new chunk needs to be accessed, but at least to some degree also while within-chunk elements are being processed.

We also attempted to decode information about the ordinal position of each chunk within the larger sequence. As we have done for the preceding decoding analyses, we initially averaged trials across all other aspects, including within-chunk positions. In this analysis, we found no evidence for reliable decoding of chunk position codes. However, it is possible that chunk order information is relevant only during chunk transitions (i.e. when the next chunk needs to be activated). Such an activation pattern would be diluted when averaging across within-chunk positions. Indeed, when we attempted to decode chunk positions separately for each within-chunk position, we obtained significant clusters in the alpha-band immediately after transitions (i.e. during the preparation period of position 1) and, in the theta-band toward the end of the probe period of position 3, that is just prior to the transition (cluster-forming threshold p < 0.05, corrected significance level, p < 0.01 for both alpha and theta effects in both studies; see *Figure 7*). No significant clusters were identified at the second position of chunks. As these results were not a-priori predicted, they need to be considered with care. However, they are consistent across the two experiments and they are consistent with the plausible assumption that chunk position representations are recruited as a new chunk needs to be established, likely as cue towards the retrieval of the upcoming chunk's identity.

## Individual differences and WM capacity

When combining Experiments 1 and 2 our sample size is—for EEG research—reasonably large and therefore allows us to probe the role of individual differences in the decoding of serial-order control codes. Behavioral results had indicated larger chunk-boundary costs in RTs, errors, or retrieval times

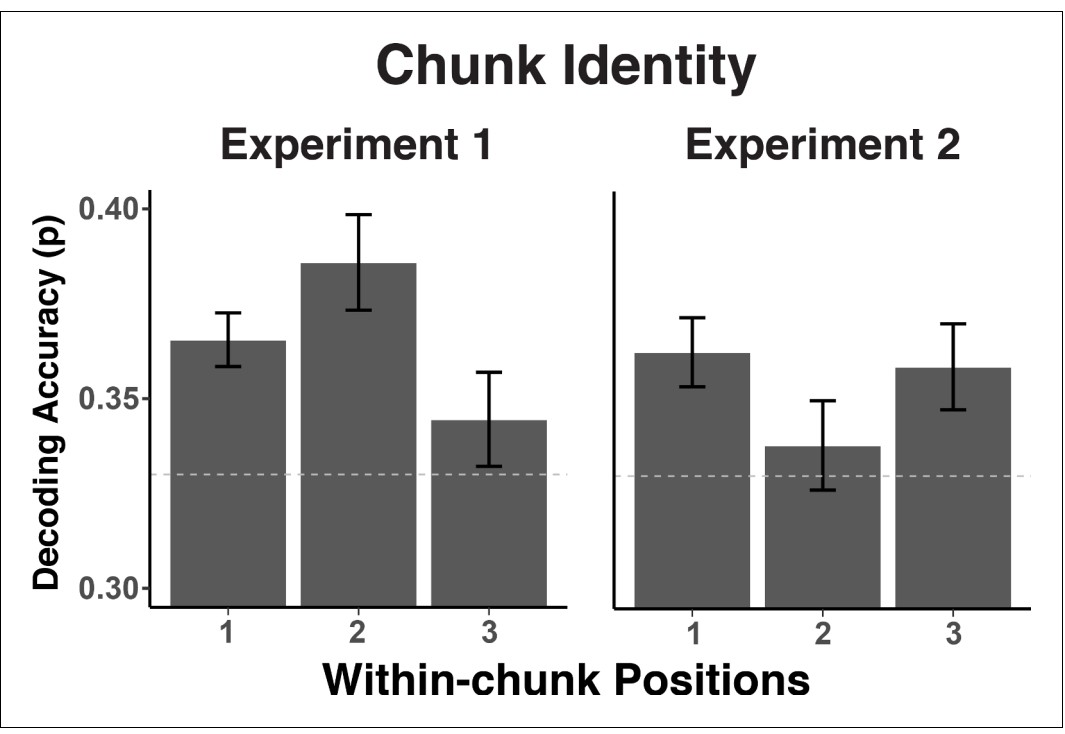

**Figure 6.** Experiment 1 and 2 decoding accuracy for chunk identity across within-chunk positions. The error bars reflect within-subject 95% confidence intervals.

DOI: https://doi.org/10.7554/eLife.38550.008

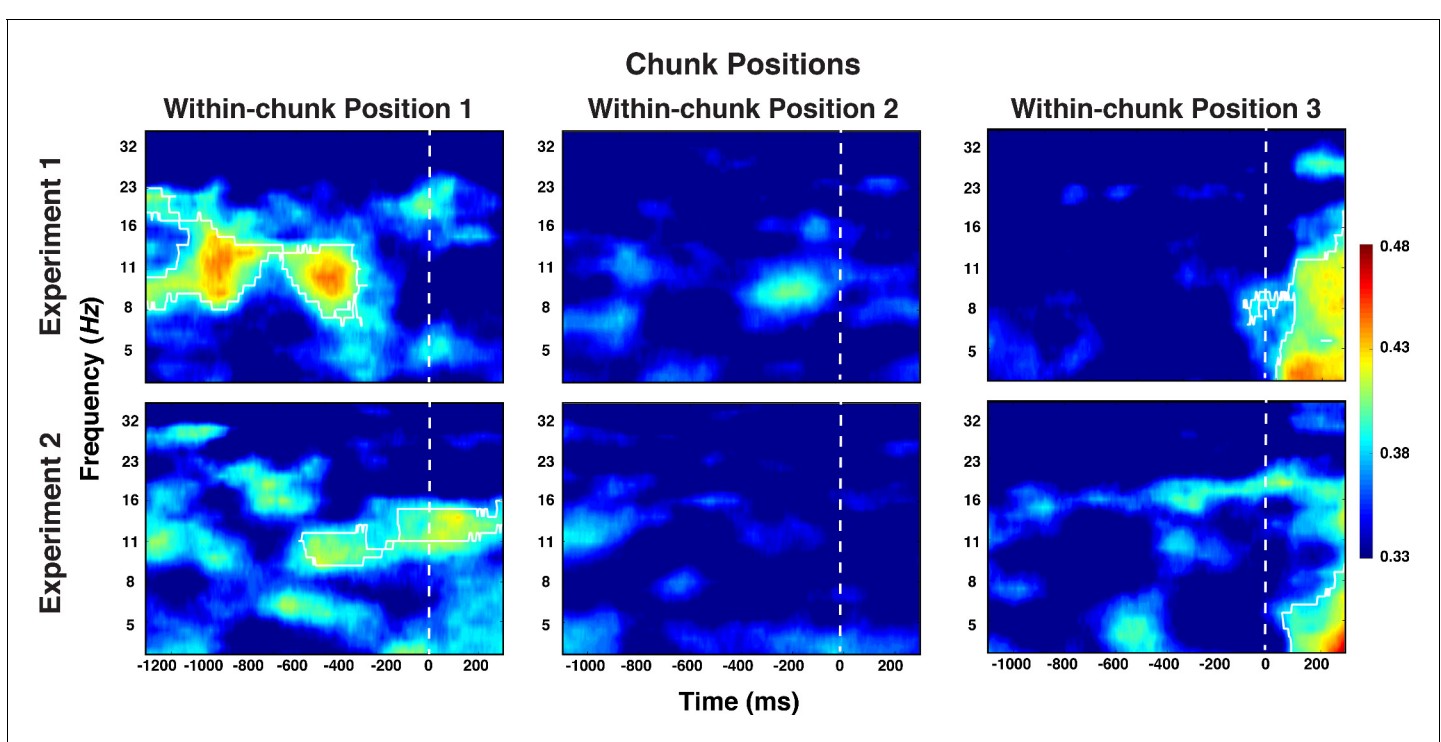

**Figure 7.** Experiment 1 and 2 decoding results as a function of frequency and time for chunk position, shown separately for the within-chunk positions 1, 2, and 3. The legend shows the decoding accuracy in probability. Chance level is p = 0.33. The regions enclosed by white lines show significant decoding accuracy after cluster correction.

DOI: https://doi.org/10.7554/eLife.38550.009

for low-WM individuals than for high-WM individuals (see *Figure 2*). This pattern is consistent with the hypothesis that WM selectively constrains the representation of level-2 codes of chunk identity and/or position. To further test this hypothesis, we examined the degree to which decoding accuracy, as an indicator of representational strength, differed as a function of WM capacity for each of the different serial-order control codes.

For Level-1 representations, we focused in the a-priori predicted frequency bands (alpha band for elements and theta band for positions). For Level-2 representations, we focused on those bands and periods (i.e. preparation vs. probe) for which we had found consistent above-chance decoding in the general analyses. As apparent from *Table 2*, neither in Experiment 1 nor in Experiment 2 did the representational strength for Level-1, basic element and position codes differ between individuals with high versus low WM capacity. For Level-2 chunk identity codes, however, decoding accuracy was robustly modulated by individuals' WM capacity during the preparation period, which was the only period during which chunk identity information was detectable. As apparent in *Figure 8*, in both experiments, there was no above-chance chunk-identity information in low-WM participants, but robust information for high-WM participants.

In order to provide a test of the differential effect of WM on the decoding of Level-2 chunk identity, but not Level-1 element identity, we also included decoding results across both experiments into an ANOVA with experiment and level as between-subject factors and level (i.e. chunk vs. element) as within-subject factor. The critical interaction between level and WM group was reliable, $F$ $(1,58) = 4.30$, $p = 0.044$, $eta^2 = 0.07$, whereas none of the interactions with the experiment factor reached significance, all $F$s < 1.

For chunk position codes, we focused on the periods at chunk beginnings (alpha band) and endings (theta band) during which reliable position decoding had been detected across the entire group. However, we found no consistent WM group differences here (see *Table 1*). Thus, overall, the pattern of WM effects on decoding accuracy was not compatible with the hypothesis that WM affects serial-order control representations in a uniform manner. Rather, WM capacity seems to selectively constrain the ability to represent the current chunk context while preparing for the next, within-chunk element.

The pattern of individual differences in decoding accuracy allows us to revisit the concern that the decoding of within-chunk positions may be driven by processing-demand confounds. The behavioral results had indicated that low-WM individuals had greater difficulty than high-WM individuals at the first chunk position (*Figure 2*), suggesting that any position-related processing-demand effects were particularly strong for that group. Thus, if position decoding is driven by differential processing demands then we would expect stronger decoding accuracy for low than for high WM individuals. However, for both Experiments 1 and 2, within-chunk position decoding accuracy was equally robust (see *Table 1*), adding to the earlier presented arguments (see *Within-Chunk Positions and Processing-Demand Confounds*) that within-chunk decoding reflects the sequential representation rather than position-specific processing demands.

## Experiment 3

A potential qualification of our position decoding results from Experiments 1 and 2 is that within-chunk sequences contained element-to-element transitions that were invariant across chunks, which in principle could allow a simple, associative chaining process to contribute to sequential representations. It is not obvious that chaining of elements can produce decoding patterns consistent with ordinal positions. Also, our direct tests of chaining-based representations revealed no corresponding evidence in Experiments 1 and 2 (see section *Evidence for Associative Chaining?*). Nevertheless, it would be reassuring to generalize our results to sequences that do not allow associative chaining. In Experiment 3, we therefore used sequences of two chunks each that were constructed from a set of six different chunks, such that for each sequence, element-to-element transitions were ambiguous and could not be learned via associative chaining (*Cohen et al., 1990*). The sequences used in this experiment optimize decoding of Level-1 element and position codes. However, with only two chunk positions and six different chunk identities they are less appropriate for decoding Level-2 codes. In particular, the fact that each sequencing block only contained two of six possible chunks introduces temporal context confounds that cannot be dissociated from chunk identity information. We therefore only focused on Level-1 element and position codes here.

**Table 2.** Average decoding accuracy (SE) for each code in individuals with high versus low working memory (WM) capacity.

| | | Experiment 1 | | | |
|---|---|---|---|---|---|
| **Level 1** | | **High WM** | **Low WM** | **T** | **P** |
| Element | Prep | 0.425 (0.03) | 0.412 (0.02) | 0.329 | 0.744 |
| | Probe | 0.427 (0.04) | 0.463 (0.02) | −0.816 | 0.421 |
| Position | Prep | 0.443 (0.03) | 0.388 (0.03) | 1.408 | 0.170 |
| | Probe | 0.477 (0.02) | 0.411 (0.03) | 1.530 | 0.137 |
| Level2 | | | | | |
| Chunk | Prep | **0.380 (0.02)** | **0.325 (0.01)** | **2.503** | **0.018** |
| | Probe | 0.368 (0.03) | 0.323 (0.04) | 0.910 | 0.370 |
| Position 3 | Prep | 0.322 (0.02) | 0.354 (0.02) | −1.107 | 0.277 |
| | Probe | 0.422 (0.04) | 0.357 (0.04) | 1.196 | 0.241 |
| Position 1 | Prep | 0.360 (0.02) | 0.371 (0.02) | 0.380 | 0.707 |
| | Probe | 0.331 (0.03) | 0.354 (0.03) | −0.597 | 0.553 |
| | | Experiment 2 | | | |
| Level 1 | | High WM | Low WM | t | p |
| Element | Prep | 0.390 (0.02) | 0.407 (0.03) | −0.403 | 0.689 |
| | Probe | 0.445 (0.04) | 0.435 (0.03) | 0.207 | 0.837 |
| Position | Prep | 0.422 (0.02) | 0.433 (0.02) | −0.380 | 0.701 |
| | Probe | 0.468 (0.04) | 0.479 (0.02) | −0.231 | 0.818 |
| Level2 | | | | | |
| Chunk | Prep | **0.407 (0.01)** | **0.326 (0.02)** | **3.427** | **0.002** |
| | Probe | 0.373 (0.03) | 0.337 (0.02) | 1.045 | 0.305 |
| Position 3 | Prep | 0.349 (0.02) | 0.340 (0.01) | 0.330 | 0.744 |
| | Probe | 0.384 (0.03) | 0.377 (0.03) | 0.162 | 0.872 |
| Position 1 | Prep | 0.348 (0.02) | 0.374 (0.03) | −0.321 | 0.750 |
| | Probe | 0.358 (0.03) | 0.372 (0.03) | −0.321 | 0.750 |

*Note.* For each feature, decoding accuracy was averaged for either the preparation or the probe period across time (i.e. −600–0 ms for the preparatory interval, 0–300 ms for the probe interval) and frequency values within theta (4–7 Hz) and alpha (8–12 Hz) bands. For the Level-2, chunk position code the difference between WM groups was tested for the observed theta-band effect in the position-3 probe period (late in the transition) and for the alpha-band effect at the position-1 preparation period (early in the transition, see *Figure 7*).
DOI: https://doi.org/10.7554/eLife.38550.012

The behavioral results (*Figure 9*) show that chunk-boundary effects on RTs were with 10.73 ms (SD = 8.5) small, albeit reliable, $F(1,19) = 19.48$, $MSE = 969.88$, p < 0.001. Also, for errors the chunk boundary effect was with 1.12% (*SD* = 3.13) considerably smaller than in the preceding experiments, albeit still reliable, $F(1,19) = 5.33$, $MSE < 0.01$, p < 0.001. Different from the self-pacing procedure in Experiment 2, we achieved the relatively small boundary effects by allowing repeatable pre-practice with each sequence. Thus, this experiment provides yet another opportunity to examine representations of ordinal positions, while minimizing potential processing-load confounds.

The decoding results fully replicated the findings for Level-1 codes from Experiments 1 and 2 (*Figure 10a*). Alpha-band activity during the preparation period and the probe period contained robust information about the specific identity of basic elements (i.e. orientations; cluster-forming threshold p < 0.05, corrected significance level p < 0.01). Ordinal positions were also successfully decoded in the theta band both for the preparation and the probe period, and to a lesser degree also in the alpha band during the preparation period (cluster-forming threshold p < 0.05, corrected significance level p < 0.01).

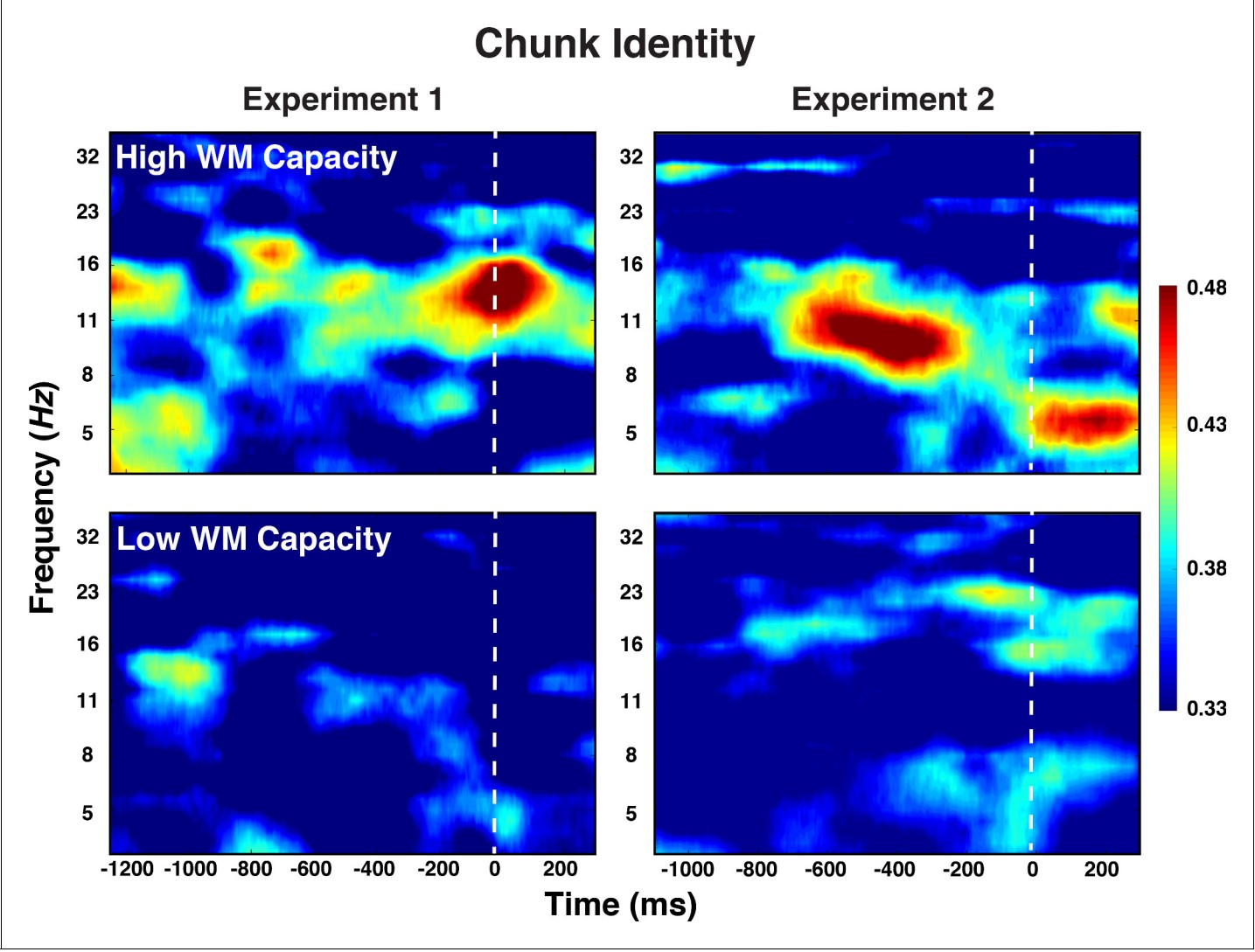

**Figure 8.** Experiment 1 and 2 decoding results as a function of frequency and time for chunk identity for individuals with low versus high WM capacity.
DOI: https://doi.org/10.7554/eLife.38550.010

We repeated the generalization analyses for within-chunk position codes that we conducted for Experiments 1 and 2. Because chunk identity was not confounded with element-position codes in Experiment 3, we were able to examine here generalization not only across different instances of elements and chunk positions, but also across chunk identities. As shown in *Table 3*, generalization scores were again very similar to decoding accuracy achieved for the reference scores. As in Experiments 1 and 2, these results are consistent with the interpretation that position codes are independent of other control codes that organize sequential performance.

As in the previous experiments, we performed RSA analyses to evaluate the possible influence of retrieval confounds on the decoding of positions (*Figure 10c*). Consistent with the results of Experiment 1 and 2, the pattern of theta activity was distinct across all three within-chunk positions during the probe period, unique-position-1: $t = 0.14$, $b = 0.04$ $SE = 0.14$; discrete: $t = 5.14$ $b = 1.24$, $SE = 0.24$. However, we found such a pattern also for the preparation period, for which we had observed a greater degree of confusion between the first position and the remaining positions in Experiments 1 and 2, unique-position-1: $t = 1.57$, $b = 0.45$ $SE = 0.29$; discrete: $t = 4.92$, $b = 0.94$, $SE = 0.19$. The additional practice with each sequence, or the need for more robust sequential representations in the face of greater between-chunk interference (due to the use of six rather than three chunks), may have induced participants to engage in a greater degree of preparation in

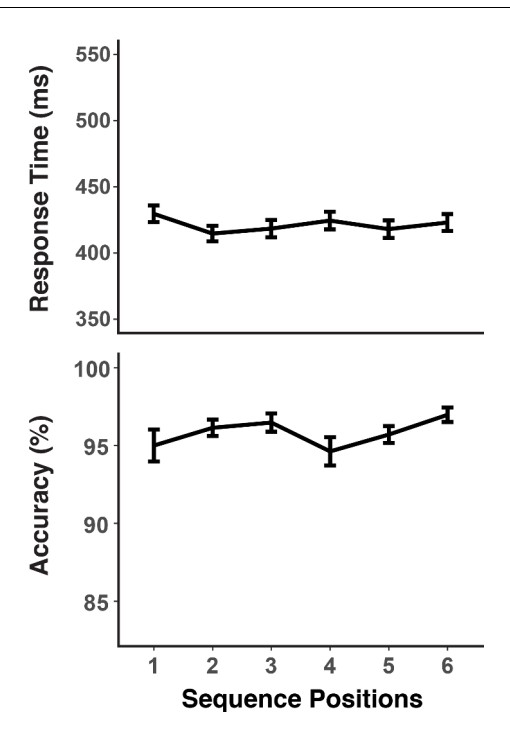

**Figure 9.** RTs and accuracies as a function of sequence positions. Error bars show within-subject 95% confidence intervals.

DOI: https://doi.org/10.7554/eLife.38550.011

Experiment 3. Combined, these results indicate that reliable position decoding is not limited to 'chainable' sequences (as used in Experiments 1 and 2) and they provide further evidence that position decoding cannot be explained in terms of processing/retrieval demand confounds.

Again, as in Experiments 1 and 2, we used multi-level regression analyses to relate trial-by-trial variability in decoding accuracy for element and position codes to trial-by-trial variability in RTs. We found that higher decoding accuracy of within-chunk position codes in the theta-band predicted faster RTs during the preparation period, $b = -0.009$, $SE = 0.0047$, $t(30) = -2.09$, but not in the probe period, $b = -0.003$, $SE = 0.006$, $t(30) = -0.51$. For element codes in the alpha-band, effects were close to reliable for the probe period, $b = -0.005$, $SE = 0.0044$, $t(30) = -1.64$, but not the probe period, $t(30) = 0.27$, $b = 0.002$, $SE = 0.007$

## Additional RSA analyses

In this experiment, the use of sequences that contained no unique element-to-element transitions or confounds between chunks and position-element conjunctions allowed us to conduct additional RSA analyses to explicitly test the independence between serial-order control codes and the plausibility of associative chaining.

## Independence of position and element codes?

The hypothesis that position and element information is independently activated during sequential performance implies that representations of positions do not depend on which element is currently relevant, and vice versa. The independence of position and element representations can be tested by analyzing the confusion matrix generated when decoding the nine combinations of each position and element. Specifically, we tested to what degree this pattern of decoding results can be accounted for by the across-trial variation in positions, the variation in elements, or the variation in position/element conjunctions (see *Figure 11a* and Representational Similarity Analysis in the Materials and methods). *Figure 11b* shows the group-average coefficients for each of the three model predictors, both for the theta and the alpha band, and both for the preparatory (600 ms interval before stimulus onset) and the probe interval (300 ms interval after stimulus onset). As shown in *Figure 11b* and consistent with the decoding results in *Figure 10a and b*, there was robust position coding in the theta band during the preparation and probe period, and element coding in the probe period. However, there was no evidence for position-element conjunctions in the theta band in the preparation and probe period. In the alpha band, the confusion matrix was accounted for by position and element codes during the preparatory period, whereas in the probe period, there was a reliable effect for element codes, and a trend for position-element conjunctions. Thus, the theta-band activity represents position information independently of element information. In the alpha band, and only during the probe period, there may be some information about how specific positions and elements are bound together.

## Integrated representations of successive elements?

A key tenet of complex chaining models based on recurrent networks is the emergence of representations that integrate successive elements (*Botvinick and Plaut, 2004*). In the sequences used in

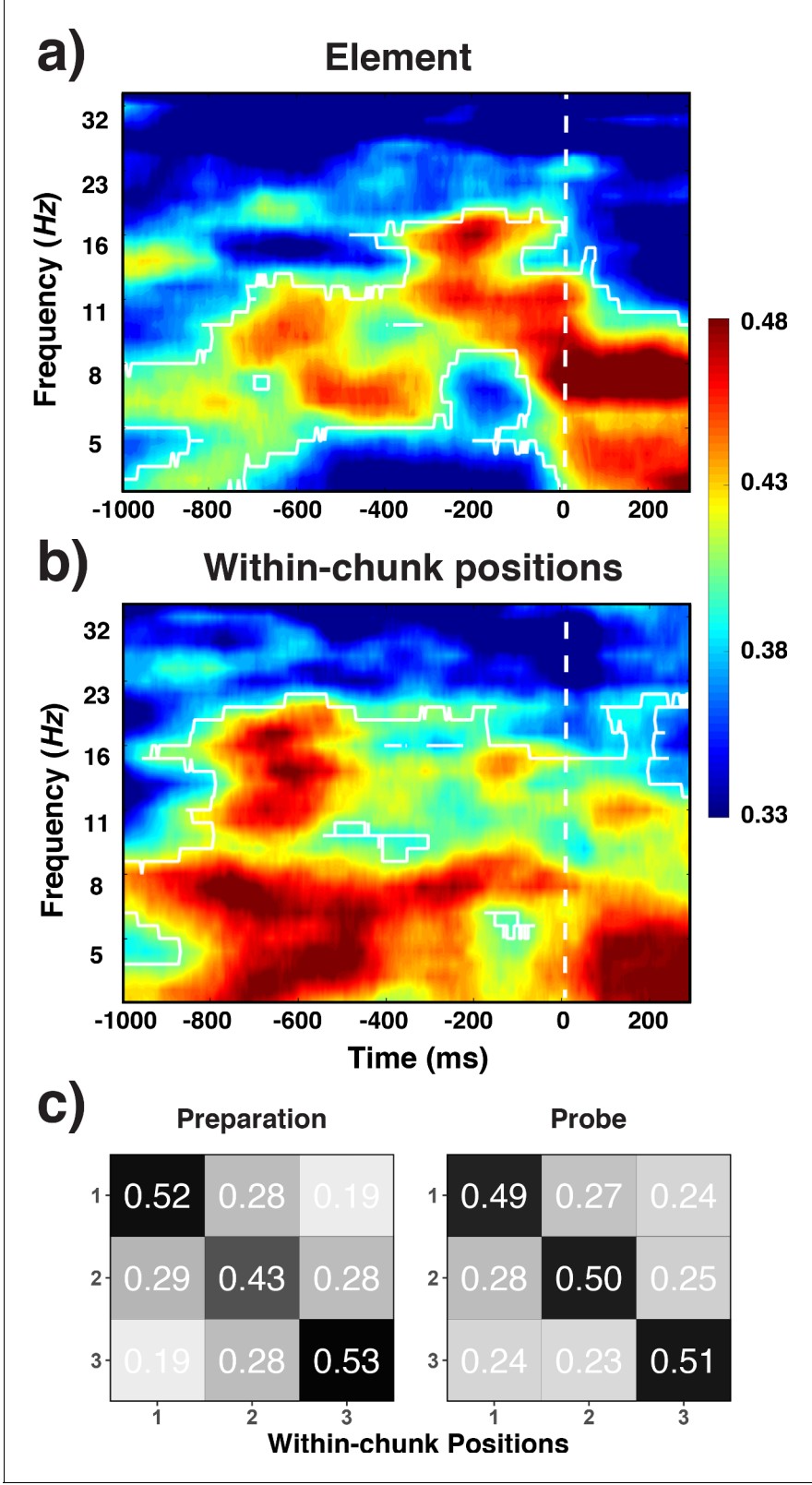

**Figure 10.** Decoding results for elements (**a**) within-chunk positions (**b**) and the confusion matrices for witin-chunk positions (**c**). Experiment 3 decoding results as a function of frequency and time for element/orientation identity (**a**) and within-chunk position (**b**). The legend shows the decoding accuracy in probability. Chance level is p = 0.33. The regions enclosed by white lines show significant decoding accuracy after cluster correction. (**b**) Confusion matrices for position decoding results from Experiment 3, separately for the preparation and the probe period.

DOI: https://doi.org/10.7554/eLife.38550.013

**Table 3.** Average generalization scores (SE) for the decoding of within-chunk position codes.

| | Preparation | | | |
| --- | --- | --- | --- | --- |
| | **Theta** | | **Alpha** | |
| Generalization variable | Reference | Generalized | Reference | Generalized |
| Elements | 0.374 (0.004) | 0.374 (0.004) | 0.373 (0.002) | 0.363 (0.002) |
| Chunk positions | 0.422 (0.007) | 0.415 (0.007) | 0.395 (0.007) | 0.393 (0.007) |
| Chunk identity | 0.392 (0.003) | 0.389 (0.003) | 0.381 (0.003) | 0.374 (0.003) |
| | Probe | | | |
| | Theta | | Alpha | |
| Generalization variable | Reference | Generalized | Reference | Generalized |
| Elements | 0.392 (0.006) | 0.383 (0.006) | 0.371 (0.007) | 0.351 (0.007) |
| Chunk positions | 0.401 (0.008) | 0.390 (0.008) | 0.378 (0.008) | 0.375 (0.008) |
| Chunk identity | 0.393 (0.003) | 0.384 (0.003) | 0.367 (0.004) | 0.353 (0.004) |

*Note.* The table shows for each generalization variable, position-code generalization scores and corresponding reference scores. Results were averaged for either the preparation or the probe period across time (i.e. −600–0 ms for the preparatory interval, 0–300 ms for the probe interval) and frequency values within theta (4–7 *Hz*) and alpha (8–12 *Hz*) bands.

Figure Captions

DOI: https://doi.org/10.7554/eLife.38550.014

Experiment 3, conjunctions between element *n*–1 and element *n* can be used to uniquely predict element *n* + 1. In principle, it is possible that participants solve the serial-order problem by representing such cross-lag conjunctions, instead of relying on position codes. We therefore examined to what degree representations of the current element (*n*) are shaped in a 'conjunctive', that is non-additive manner, by the previous (*n*-1) element. We used the EEG signal to decode combinations between element *n*-1 and element *n*. We then analyzed the confusion matrix with predictors that represented either pure element *n*-1 coding, pure element *n* coding, or the coding of conjunctions between element *n*-1 and element *n* (see *Figure 12a*). As obvious from *Figure 12b*, in no frequency band or trial phase was there any evidence for the representation of either *n*–1 elements or for conjunctions between *n*–1 and *n* elements. Thus, these results are not consistent with serial-order control through representations based on conjunctions of successive elements, as would be predicted by recurrent-network models.

## Discussion

According to a long-standing theoretical tradition, sequential behavior is organized in terms of hierarchically structured representations that are independent of the actual sequence content (*Lashley, 1951*). In principle, however, what appears to be hierarchically organized behavior can also be produced through associative chaining, where the sequential knowledge resides in associations between successive elements or complex, 'hidden-layer' integrations of such elements (*Botvinick and Plaut, 2004*). Chaining-based representations require neither content-independent codes nor an explicit, hierarchical structure.

Our results present strong evidence that is consistent with the notion that explicitly instructed, sequential action is in fact controlled via hierarchically organized, abstract codes. Evidence for sequence-extraneous control codes was particularly convincing for representations of the positions of basic sequential elements (Level-1 position codes). In addition, we found direct evidence for neural signals that represent the identity of chunks and potentially even signals that coded the position of chunks within the larger sequence. Our results complement findings from both animal neurophysiological work and human fMRI studies that have provided evidence for abstract position codes and for their neuroanatomical substrate (*Averbeck and Lee, 2007*; *Berdyyeva and Olson, 2010*; *Desrochers et al., 2015*; *Fujii and Graybiel, 2003*; *Kalm and Norris, 2014*; *Ninokura et al., 2003*). The combination of EEG decoding and an explicit sequencing paradigm allowed us to track control codes on different hierarchical levels in a time-resolved manner, while avoiding confounds that pose

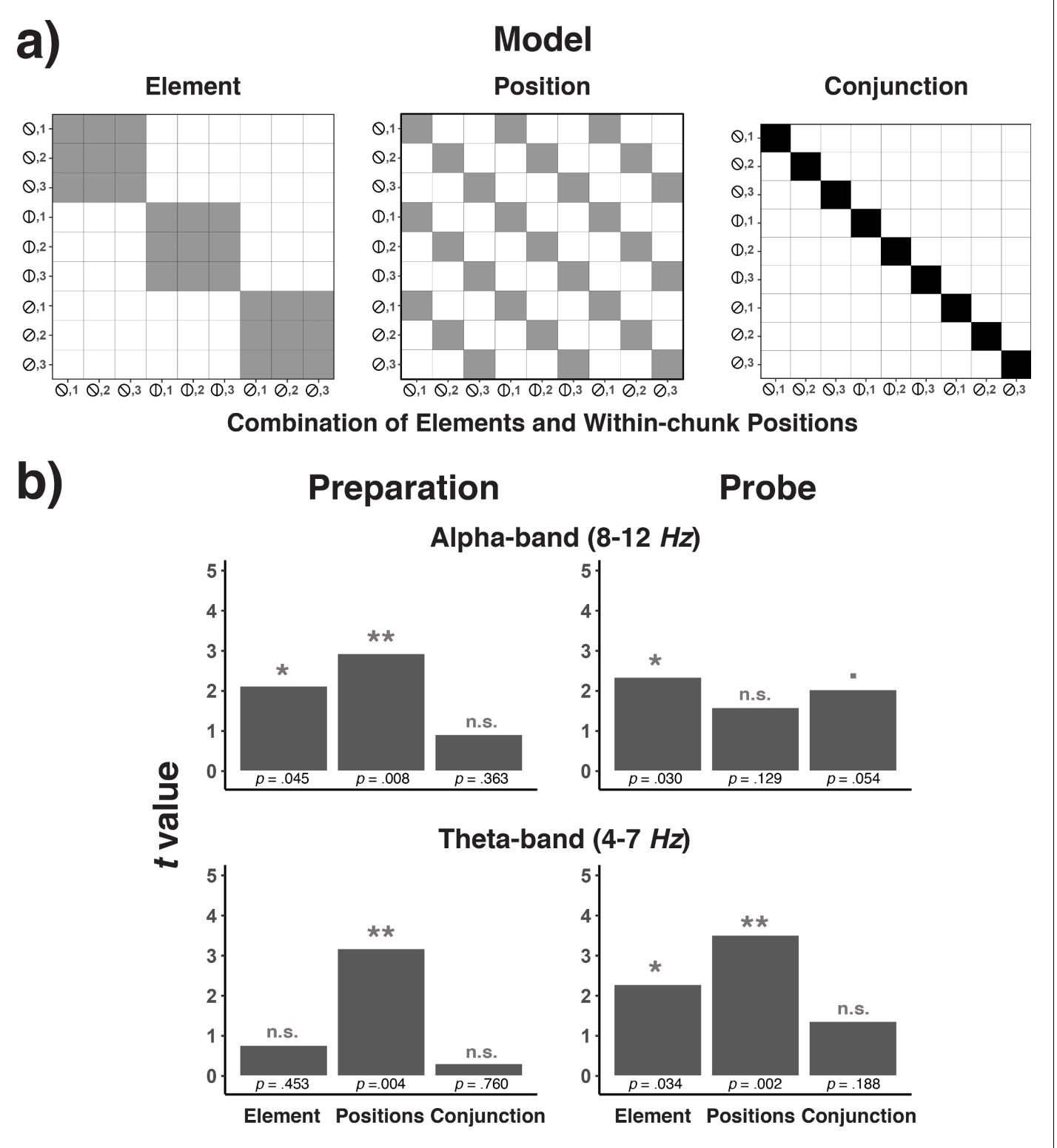

**Figure 11.** Model confusion matrices for testing element, position, and element-position conjunctions (a) and the results of testing each of these matrices across frequency bands and trial phases (b). (a) Model matrices that describe the similarity structure by the variation in elements, positions, and conjunctions of elements and positions (see the text for details). Black cells in the model matrices represent a theoretically expected correct classification probability of p = 1.0, gray cells of p = 0.33, and white cells of p = 0. (b) T-values associated with regression coefficients for each of the three model predictors are presented separately for the preparation and the probe period and for the theta- and alpha-band.
DOI: https://doi.org/10.7554/eLife.38550.015

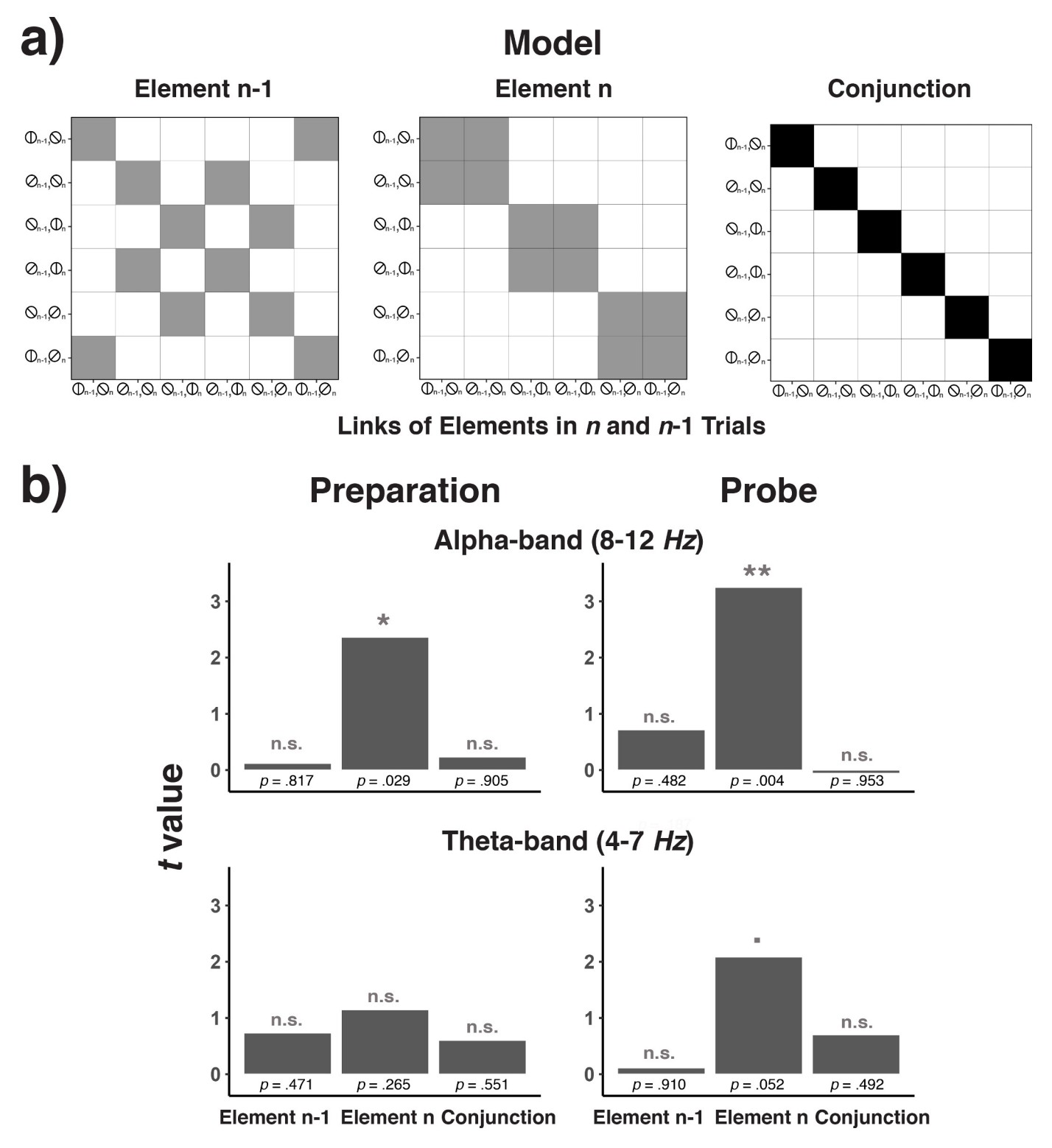

**Figure 12.** Model matrices for testing trial n-1 element codes, trial n element codes, and conjunctions between trial n-1 and trial n codes (**a**) and the results of testing each of these matrices across frequency bands and trial phases (**b**). (**a**) Model matrices that describe the similarity structure by the variation in trial *n*-1 elements, trial *n* elements, or the conjunction between trial *n*-1 and *n* elements (see the text for details). Black cells in the model matrices represent a theoretically expected correct classification probability of p = 1.0, gray cells of p = 0.5, and white cells of p = 0. (**b**) T-values

*Figure 12 continued on next page*

*Figure 12 continued*

associated with regression coefficients for each of the three model predictors are shown separately for the preparation and the probe period and for the theta- and alpha-band.

DOI: https://doi.org/10.7554/eLife.38550.016

validity challenges to position-code results in some of the earlier fMRI work (*Kalm and Norris, 2017a*; *Kalm and Norris, 2017b*).

## Position and element codes

Based on previous evidence, we had expected that basic sequential elements (i.e., different orientations) would be captured predominantly in alpha-band activity (*Foster et al., 2016*; *Fukuda et al., 2016*; *Palva et al., 2010*; *Sauseng et al., 2009*); *Foster et al. (2016)*; *Fukuda et al., 2016*), whereas within-chunk position codes would be represented in theta-band activity (*Heusser et al., 2016*; *Hsieh et al., 2011*; *Hsieh et al., 2014*).

As expected, information about the basic elements (orientations) was indeed expressed strongest in the alpha band during the preparation period, whereas information about the within-chunk positions of basic elements was expressed strongest in the theta band, both in the preparation period and the probe period. These results are the first to show within humans, that theta activity not only responds to serial-memory load (*Hsieh et al., 2011*), but also actually contains abstract, content-independent position information. In particular, detailed analyses of the representational similarity structure among the position codes provided clear evidence for relatively distinct, ordinal position codes during the probe period. In contrast, in Experiments 1 and 2, for the preparation period, the difference between position 1 versus positions 2 and 3 was particularly dominant. As elaborated in the Results section, we have some reasons to assume that this reflects the nature of the representation used during preparation. However, we cannot completely rule out the contribution of processing-demand confounds in this case. We did, however find evidence for ordinal position coding for both the preparation and the probe period in Experiment 3, where additional practice may have produced more robust representations. Overall, the finding that ordinal position information is encoded in the theta band is consistent with Lashley's original idea that serial order can be established through content-independent, position codes.

In general, there was more a quantitative than a strict qualitative dissociation between the representation of the identity of basic elements and serial-order information. For example, across all three experiments we found in the alpha band an unexpected, short phase of position decoding around the beginning of the preparation period (see *Figures 3* and *10*). The temporal signature of the decoded information at the beginning of the preparation period may indicate that it reflects the effort of activating the position code from LTM. Instead, the more sustained, theta-based activity likely represents the established position context. Conversely, during the probe period, we observed a strong presence of element information in the theta band—in the absence of any alpha-band element decoding. Indeed, the decoding of low-level spatial features of visually presented items has been previously found in stimulus-evoked theta-band oscillations (e.g. *Foster et al., 2016*). Thus, this activity may reflect the fact that bottom-up triggered representations interact with the current sequential context.

## Chunk-level representations

Larger sequences can only be represented through small sets of position codes when they are broken down into manageable chunks. These chunks in turn, need to be represented and organized in terms of their serial order. Our results provide initial evidence that just as for within-chunk information, there were distinct representations for identity and serial order at the chunk level. While chunk identity information was represented in the alpha band, there was some indication that order information was carried also by theta band activity. This pattern may suggest a more general dissociation between the representation of *what* needs to be done (elements or chunks) and *when* it needs to be done (positions of elements or chunks). It also indicates recursivity in terms of the principles by which information is represented across levels (*Fitch and Martins, 2014*).

There is a considerable body of fMRI imaging work that maps out the neuroanatomy of different levels of the cognitive control architecture (*Badre, 2008*; *Farooqui et al., 2012*; *Koechlin and Summerfield, 2007*). While our EEG-based results do not provide information about the neuroanatomcial localization of representatios on different levels, it provides information about the temporal flow of control within a hierarchical structure. One important question in this regard is to what degree higher-level representations (i.e. chunks and chunk positions) and low-level representations (i.e. elements and element positions) are activated sequentially, such that the first are needed to initiate the latter. Alternatively, higher level and lower level representations could be activated in parallel, potentially constraining each other, as (*Ranti et al., 2015*) have shown in the context of a paradigm where changes on different hierarchical levels were probed through explicit cues, rather than sequential context. Our results indicate that information about the position of a chunk within the larger sequence is activated only at chunk transition points, presumably as a cue to retrieve the next chunk identity. In contrast, information about chunks themselves was activated somewhat in parallel to information about the next orientation and its ordinal position for within-chunk element. This latter result is more consistent with the parallel-activation account.

## Working memory constraints

Individual differences in WM capacity predict peoples' ability to perform complex, sequential tasks (e.g. *Carpenter et al., 1990*). Yet, there is little research that directly examines which aspects of serial-order control are affected by WM constraints. In principle, such constraints might affect all involved representations in a uniform manner. Alternatively, constraints might be specific to particular representations. For example, according to one theoretical conceptualization (*Oberauer, 2009*), WM provides a protected space for binding a limited number of elements to a structural frame (e.g. objects to spatial locations or sequential positions). Assuming that WM is used to tie basic elements to position codes, one would expect little effect of WM capacity on these representations as long as the size of chunks does not exceed WM capacity. In fact, across two experiments we found that individual differences in WM capacity were not systematically related to Level-1 element or position codes. However, hierarchical sequences also require that WM maintains the larger context, such as of the current chunk or chunk position within the overall sequence. On the behavioral level, we found that individual differences in WM were particularly strong for between-chunk transitions, suggesting that individuals with less WM capacity had a harder time 'finding' the next chunk. The EEG results suggested that this may be due to the fact that only high-capacity individuals were able to execute within-chunk elements, while maintaining a robust representation of the current chunk identity. Thus, at least for short sub-sequences, WM constraints are specific to Level-2 chunk representations, which in turn are critical to maintain one's position within a larger sequential context. Further, these results also indicate that the answer to the question to what degree Level-2 representations are active in parallel to Level-1 representations comes with some additional nuance. At least in our paradigm, the exact architecture of control seems to depend on available WM capacity, where only high-capacity individuals exhibit strong evidence for parallel Level-1 and Level-2 activity.

On the basis of our results, we cannot conclude that there is an unconditional dissociation between Level-1 and Level-2 control codes and the relevance of WM constraints. Specifically, it is possible that WM capacity constrains Level-1 representations when the primary sequential elements have a larger representational load, such as when using tasks with arbitrary S-R mappings (*Kikumoto and Mayr, 2017*). Equally, WM capacity should become relevant with larger chunks (i. e. >3) that may fit into the workspace available to high-WM individuals, but that may have to be broken up into smaller chunks in low-WM individuals (*Bo and Seidler, 2009*). Thus, it will be important to replicate our results across a broader range of sequential elements and sequencing demands.

## Position codes versus chaining

We used a paradigm that was geared towards establishing robust evidence for abstract, serial-order control codes—should such codes indeed exist. In particular, participants had to work through a large number of relatively short blocks, each of which required adopting new, explicitly instructed sequences. This procedure should have made it more likely for participants to rely on flexible, content-independent serial-order control codes than on associative links between consecutive elements. Also, past behavioral work using a similar paradigm has produced strong evidence that participants

did in fact rely on content-independent position codes (e.g. *Mayr, 2009*; see in particular Experiment 5). Indeed, several of our analyses supported the notion of independent position codes. For example, within-chunk position codes showed robust generalization across other aspects such as elements, chunk position, and chunk identity (the latter only tested in Experiment 3). In Experiment 3, we also explicitly tested to what degree position information and element information were coded independently or in terms of integrated position/element representations. Except for a non-significant trend during the probe period in the alpha band, there was no evidence for integrated representations.

We also conducted two additional, explicit tests of associative chaining. In Experiments 1 and 2, the makeup of our sequences would have allowed very simple, chaining-based representations to emerge. *Schapiro et al. (2012)*, had proposed to examine to what degree element n-1 representations become more similar to element n representations through experience as an indicator of associative chaining. This analysis provided no indication that associative chaining was a significant factor. In Experiment 3, we conducted RSA analyses to probe the existence of representations that combine unique combinations of consecutive elements in an integrated manner. Such integrated representations are a critical component of complex chaining models (*Botvinick and Plaut, 2004*) and would be a necessary condition for chaining-based serial-order control with the sequences used in Experiment 3. Our analyses demonstrated robust decoding of the current element, but no evidence for decoding of the previous element, or of the conjunction of the current and the previous elements.

While our results provide positive evidence for the existence of content-independent position codes, they cannot be used to rule out that associative chaining representations govern behavior in different types of sequencing situations. In particular, we might expect associative chaining to emerge after extensive practice with a particular sequence, or when position information is not informative (*Botvinick and Plaut, 2004*; *Davachi and DuBrow, 2015*; *Keele et al., 2003*). In future work, the same types of analyses we used here to probe the existence position codes, should also be useful for tracing the emergence of chaining-based representations.

## Qualifications

One important qualification is that our procedures were particularly geared toward a robust decoding of within-chunk elements and position codes by presenting each element and each position combination across the three chunks in each sequence. Had we assessed each chunk in separate blocks, decoding would have been affected by common temporal context, likely making it more difficult to extract pure position-specific or element-specific information. While the use of three chunks per sequence in Experiments 1 and 2 also allowed us to decode Level-2 representations, the design was less geared toward the robust assessment of chunk identify or order information. In particular, the results regarding Level-2 chunk position codes need to be considered with some caution. The fact that chunk position decoding was limited to the boundaries between chunks (i.e. for chunks at the end of third position and the beginning of first position) is theoretically plausible. However, it also makes it particularly difficult to rule out retrieval or other processing demands that may be confounded with chunk positions. Thus, whether the information decoded at chunk boundaries is indicative of a Level-2 control code, or of the system's attempt to generate such a code (i.e. through retrieval) cannot be addressed conclusively in the current work.

An obvious limitation of our work is that while the EEG-based decoding approach yields high temporal resolution, it provides only little information about the neuroanatomical origin of the observed rhythmic activity (see Appendix). In this regard, previous research suggests some hypotheses. In serial memory tasks, a frontally distributed enhancement of theta power has been observed that is commonly associated with the medial and lateral PFC (*Hsieh et al., 2011*; *Jensen and Lisman, 2005*; *Jensen and Tesche, 2002*). This finding converges with monkey electrophysiology (*Averbeck and Lee, 2007*; *Berdyyeva and Olson, 2010*) and human fMRI studies (*Desrochers et al., 2015*; *Kalm and Norris, 2017b*; *Ninokura et al., 2003*) in which neural activity in the PFC showed sensitivity to both serial positions and hierarchical control demands (*Badre and D'Esposito, 2009*). Furthermore, the hippocampus, where theta oscillations are prominent, is likely a critical structure for coding temporal context (*Heusser et al., 2016*; *Hyman et al., 2005*; *Jones and Wilson, 2005*; *Siapas et al., 2005*). More specifically, it is possible that the hippocampus serves as a driver of prefrontal theta modulation (*Ferino et al., 1987*). To better integrate

anatomical and temporal information, the cycling-sequence paradigm used here can provide a solid basis for future work that combines fMRI and EEG (or MEG) decoding of serial-order representations in the context of the same, well-understood model task (*Desrochers et al., 2015*; *Mayr, 2009*; *Schneider and Logan, 2006*).

## Conclusion

Our results suggest that time-resolved decoding analyses with EEG can be utilized to characterize abstract control representations. Specifically, we provide strong evidence that oscillatory activity in the theta band contains information about sequential position codes that is independent of the representation of the content of a sequence. In addition, we provide initial evidence about the representation of chunk-level information, as well as about how WM constraints affect representations of chunk identity.

# Materials and methods

## Participants

A total of 88 students of the University of Oregon participated in this research and received compensation of $10 per hour. Four participants from Experiment 1, two participants from Experiment 2, and two participants from Experiment three were excluded because of a higher than 25% trial rejection rate due to EEG artifacts. The final samples contained 30 participants each for Experiments 1 and 2, and 20 participants for Experiment 3. All experimental protocols and procedures were approved by the University of Oregon's Human Subjects Committee. Past work has reported robust EEG decoding of perceptual or working memory representations with sample sizes of less than 20 participants (e.g. *Foster et al., 2016*). Given that this is the first study to look at the decoding of abstract control representations, we tried to establish the robustness of our key findings by both using larger samples for two of our experiments ($N$ = 30 in Experiments 1 and 2) and by replicating key results across experiments. The larger sample sizes and consistent sequence designs across Experiments 1 and 2 also allowed us to explore individual differences in decoding accuracy by combining data across these experiments.

## Task and stimuli

On each individual trial of the experiment, participants were asked to retrieve an orientation associated with the current position within a larger pre-instructed sequence and compare it with an orientation probe stimulus, presented on the screen (*Figure 1b*). These orientations were presented as lines within a circle (2°in diameter) in which lines were tilted in 45°, 90°, or 135° angles. Each trial contained a preparation and a probe period. During the preparation period, participants were instructed to pre-retrieve the corresponding element for the upcoming test as quickly as possible and maintain the orientation until the probe arrives. In previous work, we had established that participants are able to consistently follow such instructions (*Mayr et al., 2014*). The presentation phase was indicated by the fixation cross (0.25° in diameter) presented at the center of the screen and lasted 1250 ms, 1100 ms, or 1000 ms for Experiments 1, 2, and 3, respectively. During the probe period, a potential orientation stimulus was presented, and participants indicated per key press whether or not the probe matched the element corresponding to the current position within the sequence ('z' key pressed with the left index finger for non-matches; '/' key with the right-index finger for matches). In order to encourage preparation, mild time-pressure was induced by requiring responses within a 700 ms time window; later responses were counted as errors. After the response followed a jittered, inter-trial-interval (ITI) of 500 to 900 ms. Additionally, for Experiment 2, the preparation period was preceded by a self-paced 'retrieval period', where participants initiated the preparation period by pressing the space bar in a self-paced manner, once they had retrieved the next element.

Given that sequences cycled through repeatedly within each block, it was important to ensure that participants 'got back into the sequence' following an error. Therefore, after each error, the probe stimulus remained on the screen, the correct sequence was displayed at the bottom of the screen and the element corresponding to the current position was highlighted. This error feedback was presented until the correct response was executed and it remained on the screen into the

following trial. These error and post-error trials were excluded from all analyses. Participants were encouraged to make judgments to the probe as fast and accurate as possible. All task-related visual stimuli were generated in MATLAB (Mathworks) using the Psychophysics Toolbox (*Brainard, 1997*) and were presented on a 17-inch CRT monitor (refresh rate: 60 Hz) at a viewing distance of 100 cm.

## Sequences and blocks

Prior to each experimental block of trials, a new sequence was instructed. Participants were instructed to learn each sequence in form of chunks of three orientations. Accordingly, the instruction screen displayed the sequence, spatially organized in terms of its three chunks. In Experiments 1 and 2, participants were given a new nine-element sequence made of three unique chunks of three unique elements (ABC, BCA, CAB) for each block of trials (*Figure 1a*). Counterbalancing the order of all chunks yields six unique sequences, each of which was presented equally often within 24 experimental blocks of 54 trials each (i.e. six sequence cycles per block). For Experiment 3, participants were instructed six-element sequences made of two, 3-element chunks and selected from a total of six possible chunks (ABC, ACB, CAB, CBA, BAC, BCA). In this experiment, sequences were constructed such that elements never repeated across chunk boundaries, which yields a total of 12 possible sequences. Participants performed 36 experimental blocks of 42 trials each (i.e. seven sequence cycles per block). In all three experiments, this procedure ensured that each element (orientation), each position, chunk identity, and chunk position are combined equally often across blocks.

Prior to the experimental blocks, participants were given practice blocks. In Experiments 1 and 2, each of three chunks were independently tested in the aforementioned match/mismatch task, and participants performed three practice blocks of 36 trials each (i.e. 12 chunk cycles per block). In Experiment 3, we attempted to provide more robust pretraining with individual chunks with the hope to counteract position-dependent processing demands (e.g. *Kalm and Norris, 2017a*). To this end, a first chunk was randomly selected from the pool of six possible chunks and repeatedly tested until it was reproduced with 100% accuracy by clicking on the three possible orientations in the correct order for a given chunk. At this point, the next chunk was added and tested, randomly intermixed with the first chunk until the 100% performance criterion was achieved. This procedure was repeated until the performance for all chunks reached the performance criterion. Given that each error within a practice block led to the repetition of that block, this procedure ensured considerable exposure to the full set of chunks (total duration about 20 min, compared to under 5 min for Experiments 1 and 2).

## Measurement of WM capacity

To assess participants' WM capacity, we chose a standard, visual change-detection paradigm (*Luck and Vogel, 1997*) because it imposes no serial-order demands. Participants were presented arrays of four or eight colored squares (0.65°×0.65°) at random locations during a 100 ms encoding interval. After a delay period of 900 ms, a single probe at the location of one of the squares of the encoding set was presented. Participants had to indicate whether or not the color of the probe matched the color of the original square. There were 160 trials each for set-sizes 4 and 8. The capacity was computed as $K = S \times (H-F)$, where $K$ is the memory capacity, $S$ is the size of the array, $H$ is the observed hit rate and $F$ is the false alarm rate (*Cowan, 2001*).

## EEG recordings and preprocessing

Electroencephalographic (EEG) activity was recorded from 20 tin electrodes held in place by an elastic cap (Electrocap International) using the International 10/20 system. The 10/20 sites F3, Fz, F4, T3, C3, CZ, C4, T4, P3, PZ, P4, T5, T6, O1, and O2 were used along with five nonstandard sites: OL midway between T5 and O1; OR midway between T6 and O2; PO3 midway between P3 and OL; PO4 midway between P4 and OR; and POz midway between PO3 and PO4. The left-mastoid was used as reference for all recording sites. Data were re-referenced off-line to the average of all scalp electrodes. Electrodes placed ~1 cm to the left and right of the external canthi of each eye recorded horizontal electrooculogram (EOG) to measure horizontal saccades. To detect blinks, vertical EOG was recorded from an electrode placed beneath the left eye and reference to the left mastoid. The EEG and EOG were amplified with an SA Instrumentation amplifier with a bandpass of 0.01–80 *Hz* and

were digitized at 250 *Hz* in LabView 6.1 running on a PC. We used the Signal Processing, EEGLAB (*Delorme and Makeig, 2004*) toolboxes, and custom-made codes for EEG processing in MATLAB. Trials including blinks (>80 uv, window size = 200 ms, window step = 50 ms), large eye movements (>1°, window size = 200 ms, window step = 10 ms), and blocking of signals (range = −0.05 *uv* to 0.05uv, window size = 200 ms) were excluded from further analysis, resulting in an average of 920 trials per participants.

## EEG analysis

Single-trial EEG data were decomposed into their time-frequency representation by a complex wavelet convolution. Following *Cohen (2014)*, the power spectrum of the EEG signal was obtained through fast Fourier transformation of the raw EEG signal. The power spectrum was convolved with a series of complex Morlet wavelets ($e^{2\pi ft}e^{-t2/(2*\sigma^2)}$), where *t* is time, *f* is frequency increased from 4 to 35 Hz in 32 logarithmically spaced steps, and σ defines the width of each frequency band, set according to *n/2πf*, where *n* increased from 3 to 10. The incremental number of wavelet cycles was used to balance temporal and frequency precisions as the function of frequency of the wavelet and the logarithmic scaling was used to keep the width across frequency bands approximately constant. After this is done in the frequency-domain, we took the inverse of the Fourier transform, resulting in complex signals in the time-domain. A frequency band-specific estimate was defined as the squared magnitude of the convolved signal $Z$(real([z(t)]$^2$ + imag[z(t)]2) for instantaneous power.

## Frequency-by-time decoding analysis

In order to investigate whether the pattern of oscillatory power carries information about the representations of hierarchical sequence, we performed a series of time-resolved decoding analysis via linear discriminant analysis (*Fisher, 1936*; *Grootswagers et al., 2017*; *Stokes et al., 2015*; *van Ede et al., 2017*). Each control code (i.e. elements, within-chunk positions, chunk identity, and chunk positions) was decoded to map out the spectral-temporal property of these representations by training independent decoders over the range of frequency values and time samples.

Each decoding result, which essentially corresponds to a pixel in the frequency X time decoding results (e.g. *Figure 3*), was independently evaluated by using *k*-fold cross-validation procedure (*Mosteller and Tukey, 1968*). To account for the issue of statistical multiple comparisons, we used the cluster-based permutation test (see below for detail). First, all artifact-free trials, excluding errors and post-error trials were randomly partitioned into four independent folds. Next, within each fold, trials were further grouped by each instance of the to-be-decoded variable. Then the oscillatory power was averaged over trials for each group and electrode with a 100 ms (i.e. 25 time-samples) sliding window. To prevent biasing the decoders, the number of observations for each instance was equated within and across folds by randomly dropping trials in folds with higher numbers of observations prior to the averaging. Thus, each fold consisted of an c × e × f × t matrix of power values, where *c* is the number of instances of to-be-decoded variable, *e* is the number of electrodes, *f* is the number of frequency values, and *t* is the number of time samples. Further, to remove the effects of potential processes that affect multiple electrodes in a uniform manner (e.g. sensory adaptation, *Kalm and Norris, 2017a*), all data points were transformed into z-scores across electrodes, but separately for each timepoint and individual. Then, three of the folds were used as the training set, and the remaining fold served as the test set. This process was repeated until each fold had served as the test set. To improve the signal-to-noise ratio of the decoding results, we introduced an iterative procedure in which the entire cross-validation steps described above was repeated 30 times, and each iteration generated a new set of folds with randomly assigned trials. Finally, test-set decoding results were averaged across all iterations.

## Cluster-based permutation test

Non-parametric permutation tests were used to evaluate the decoding results in the time-frequency space on the group level (*Maris and Oostenveld, 2007*). First, we identified clusters of samples with reliable decoding accuracy (i.e. *t*-value >2.0) that were adjacent in the temporal and the frequency dimension. Then, empirical cluster-level statistics were obtained by taking the sum of *t*-values in the largest cluster. Finally, nonparametric statistical tests were performed by calculating a cluster *p*-value under the permutation distribution of cluster-level statistics, which were obtained by Monte Carlo

iterations of decoding analyses with randomly shuffled condition labels. Both the cluster entry threshold and the cluster significance threshold were set to p < 0.05, one-sided. A one-sided threshold is appropriate here, as were only interested in clusters that were more accurate than chance.

For all additional analyses, such as the generalization analysis, representational similarity analysis, and multi-level modeling of RTs (see below), we modified the aforementioned cross-validation steps such that power values were averaged within a priori selected frequency bands: theta- (4–7 Hz) and alpha- (8–12 Hz) band (i.e. reducing the f dimension of a 'c × e × f × t fold' onto two categories). Then, the resulting, time-resolved decoding results were averaged within the preparation period (−600 to 0 ms interval prior to the stimulus onset) and the probe period (0 to 300 ms, post-stimulus onset). Other procedures in the cross-validation steps remained the same as described above. Because statistical tests were performed on the averaged results, we did not perform cluster-based permutation correction for these analyses.

## Generalization analysis

Generalizability of within-chunk position codes was tested by cross-classifying positions across instances of other control codes (e.g. elements). The aforementioned fourfold, repeated cross-validation steps were modified such that independent training and test sets were further subdivided by instances of a 'generalization variable' (e.g. 45°, 90°, 135° orientations). To obtain generalization scores, both the training sets and test sets were filtered depending on the corresponding instance of a generalization variable. Each test set included only one instance of the generalization variable (e.g. trials with 45° orientation), while the training sets included all other instances (e.g. trials with 90° or 135° orientation). The filtering step was repeated until all unique instances of a generalization variable served as the test sets. Because this procedure reduced the amount of training data, the decoding results are expected to become lower than the results with the full data sets. Therefore, we computed reference scores by applying the same decodes that were trained on the filtered data to the separate test sets that included one of the instances used as the criterion for the training sets (i.e. trials with 90° or 135° orientation).

## Representational similarity analysis

Representational similarity analysis (RSA) probes the representational space that gives rise to the observed, multivariate neural pattern (*Nili et al., 2014*). In the current context, RSA analyses were applied to subject-level confusion matrices from the time-resolved decoding analyses (note that *Figure 4a*, *Figure 11a*, and *Figure 12a* show group-averaged confusion matrices). Specifically, the x-axis of each matrix corresponds to the correct instance and the y-axis represents instances selected by the decoders. For example, in *Figure 4b* the left-most column of the matrix shows the relative frequencies with which each instance was selected by the decoder when the correct instance was position 1. Note, that relative frequencies within each column add up to 1.0.

These empirical matrices can be analyzed through model matrices that correspond to competing theoretical assumptions about the underlying representational structure. For example, the unique-position-1 model matrix in *Figure 4a* assumes that the representation differentiates between position one and the remaining positions 2 and 3, with no further differentiation between the latter two. If the empirical representation complies perfectly with this idealized scenario, the decoder would accurately predict 100% of correct position one cases and would evenly confuse position 2 and 3 cases (i.e. with p = 0.5), but never positions 2 or 3 with position 1. Therefore, the model matrix shown in *Figure 4a* contains p = 1.0 in the black cell: p = 0.5 in the gray cells and p = 0 in the remaining cells. For each RSA, the competing model matrices served as simultaneous predictors within a multi-level regression model, and the logit-transformed, relative frequencies in the corresponding, empirical confusion matrix as the criterion variable. Model predictors were entered as fixed effects and subject-contingent intercepts and slopes as random effects.

## Multi-level modeling

To investigate how the trial-to-trial variability in the quality of control codes relates to trial-by-trial variability in RTs, we used multi-level modeling to predict trial-level RTs by single-trial indicators of decoding accuracy for relevant control codes. Specifically, we retained the logit-transformed posterior probability of assigning the correct label for each trial and control code in the test sets. The

posterior probability can be used as a continuous variable expressing the certainty with which the correct instance is represented on that trial. We used the alpha-band information to indicate the element code and we used the theta-band to indicate the position code. These analyses were conducted separately for the preparation and the probe period. After excluding errors and post-error trials, log-transformed RTs were residualized by the linear and quadratic trends of experimental trials and blocks, contrasts for position 1 vs. positions 2 and 3 trials, and position 2 vs. position 3 trials, and match/mismatch between the current element and the probe (coded 0/1). We then regressed these residualized RTs on the posterior probability for elements in the alpha band and for positions in the theta band as fixed-effect predictors, as well as subject-contingent intercepts and slopes as random effects. We conducted these analyses separately for the preparation period ($-600$ to $0$ ms interval prior to the stimulus onset) and the probe period ($0$ to $300$ ms, post stimulus onset).

### Data access
Access to all data and analysis files is provided through Open Science Framework.

## Acknowledgements

This work was supported by NSF grant 1734264 awarded to Ulrich Mayr.

## Additional information

### Funding

| Funder | Grant reference number | Author |
|---|---|---|
| National Science Foundation | 1734264 | Ulrich Mayr |

The funders had no role in study design, data collection and interpretation, or the decision to submit the work for publication.

### Author contributions
Atsushi Kikumoto, Conceptualization, Formal analysis, Investigation, Visualization, Methodology, Writing—original draft, Writing—review and editing; Ulrich Mayr, Conceptualization, Funding acquisition, Investigation, Methodology, Writing—original draft, Project administration, Writing—review and editing

### Author ORCIDs
Atsushi Kikumoto (iD) http://orcid.org/0000-0002-2179-2700
Ulrich Mayr (iD) http://orcid.org/0000-0002-7512-4556

### Ethics
Human subjects: We obtained informed consent from human subjects. Consent and study procedures were approved by the University of Oregon's Human Subjects Institutional Review Board (Protocol 10272010.016).

### Decision letter and Author response
Decision letter https://doi.org/10.7554/eLife.38550.024
Author response https://doi.org/10.7554/eLife.38550.025

## Additional files

### Supplementary files
• Transparent reporting form
DOI: https://doi.org/10.7554/eLife.38550.017

## Data availability

All data and analysis scripts have been deposited at OSF (https://osf.io/6hmrz/).

The following dataset was generated:

| Author(s) | Year | Dataset title | Dataset URL | Database and Identifier |
|---|---|---|---|---|
| Kikumoto A, Mayr U | 2018 | Decoding Serial Order Control | https://osf.io/6hmrz/ | Open Science Framework, 6hmrz |

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

## Appendix 1

DOI: https://doi.org/10.7554/eLife.38550.018

### Topography of Decoding Results

We examined the topographical pattern associated with the decoding of element and within-chunk position codes and its consistency among individuals. To this end, classification weight vectors (i.e., coefficients) of each electrode assigned by linear decoders were transformed and projected onto the scalp as an 'activation pattern' (*Grootswagers et al., 2017*; *Haufe et al., 2014*). Specifically, the spatial pattern of activation (*A*) was reconstructed by multiplying classification weights (*w*) by the covariance of EEG data (*X*): $A = XX^{T} \times w$; where $\underline{w}$ is a weight vector of length *M* and *X* is $N \times E$ matrix of EEG data with *N* observations and *M* features (i.e., electrodes). This procedure allows to estimate the contribution of each electrode for the decoding results by mapping non-zero weight values to changes in the amount of class-specific information. In our study, there were three unique instances for elements and within-chunk positions for all experiments, thus decoders assigned coefficients of the boundary curves for each pairwise comparisons of instances (i.e., 45° vs 90°). Therefore, we computed the activation pattern by converting each set of coefficients separately and then normalized these by their vector norms.

Using the above-mentioned weight projection method, the activation patterns were separately reconstructed by running the decoding analyses with the data averaged within the preparation or the probe period (i.e., −600–0 ms for the preparatory interval, 0–300 ms for the probe interval) and frequency values within the theta (4–7 *Hz*) and the alpha band (8–12 *Hz*); we excluded the theta-band during the preparation phase which had not shown significant decoding of elements in the main results (see *Figure 3* and *Figure 10*). The inter-individual consistency of reconstructed activation patterns was tested by running a series of regression analyses within each electrode, phase, and control code. Specifically, at a given electrode, we tested whether the individual-specific, converted weights systematically deviated from 0, which would indicate that the observed EEG activity contributed to the decoding of the targeted control code in a consistent manner among individuals. The regression was separately performed for each type of pairwise comparisons among instances (e.g., 45° vs 90° and 90°. vs 135° and 45°. vs 135°). Then, resulting *t*-values were averaged among these comparisons. Finally, the topography was generated by fitting a smooth spline via a generalized additive model (*Wood, 2017*).

*Figure 1* shows the topography of *t*-value in which red represents positive correlations that are consistent across individuals between the source with the corresponding electrodes, while blue indicates negative correlations. As apparent, in most cases the group-based consistency of activation patterns was relatively weak, with very few *t*-values in the significant range and little indication of consistency across experiments. This suggests that most of the decoding results that we reported in the main text were generated through individual-specific patterns of rhythmic activity over scalp. The only across-experiment, consistent exception regards the coding of positions in the theta band during the preparation phase (and for Experiment three also in the alpha band). Here, we found negative weights in left-central electrodes (e.g., C3 and Cz) and a tendency towards positive weights in the remaining areas. For this particular case, we also present individual topographies (see *Figure 2*). These demonstrate that even in this case of relatively consistent group-level patterns, there was additional, considerable idiosyncratic variability. Given that we had no a-priori predictions about the topography of decoding weights, we refrain from post-hoc interpretations.

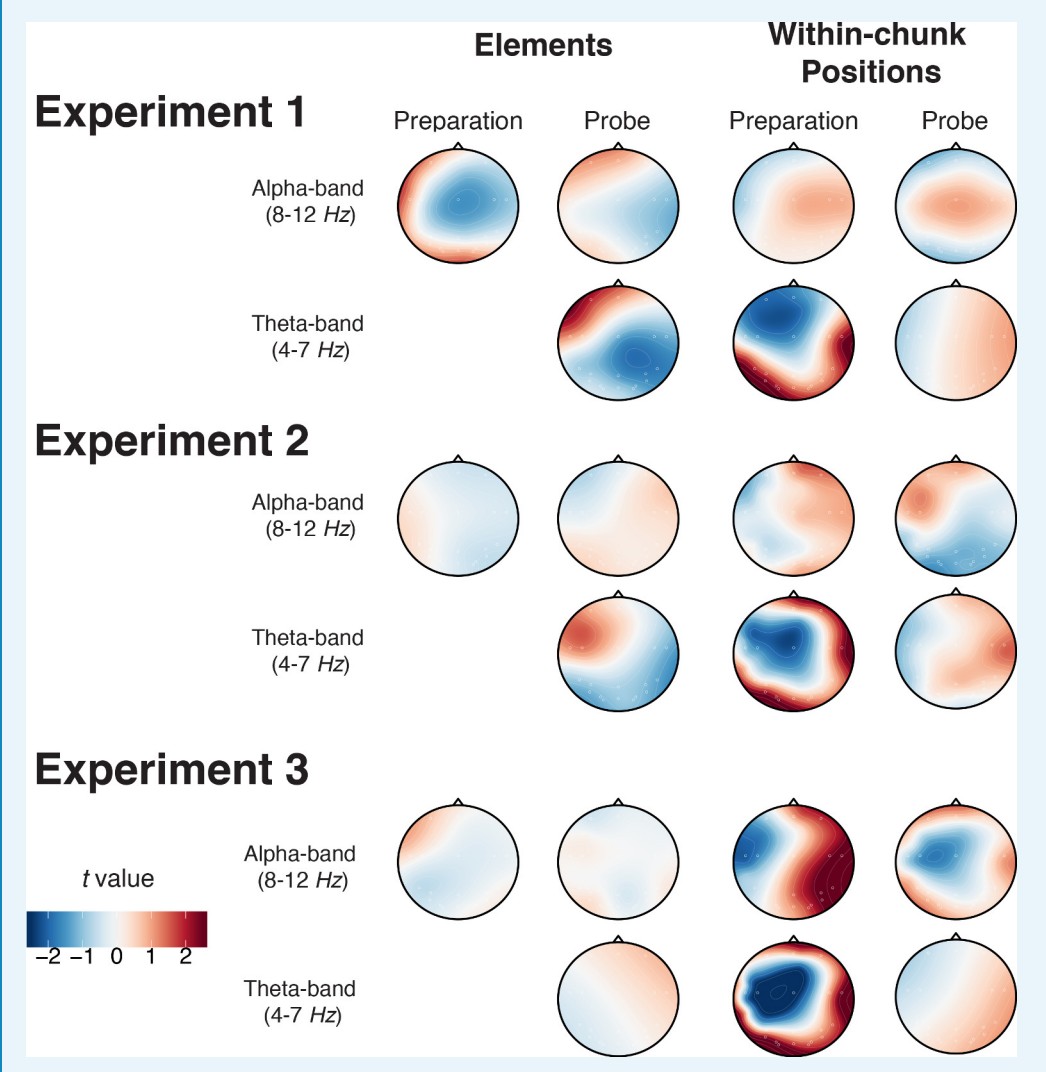

**Appendix 1—figure 1.** Topography of reconstructed activation patterns for element and within-chunk position codes across Experiment 1, 2 and 3. Significant *t*-values indicate that the EEG activity at the given electrode contributed to the decoding results consistently among individuals.

DOI: https://doi.org/10.7554/eLife.38550.019

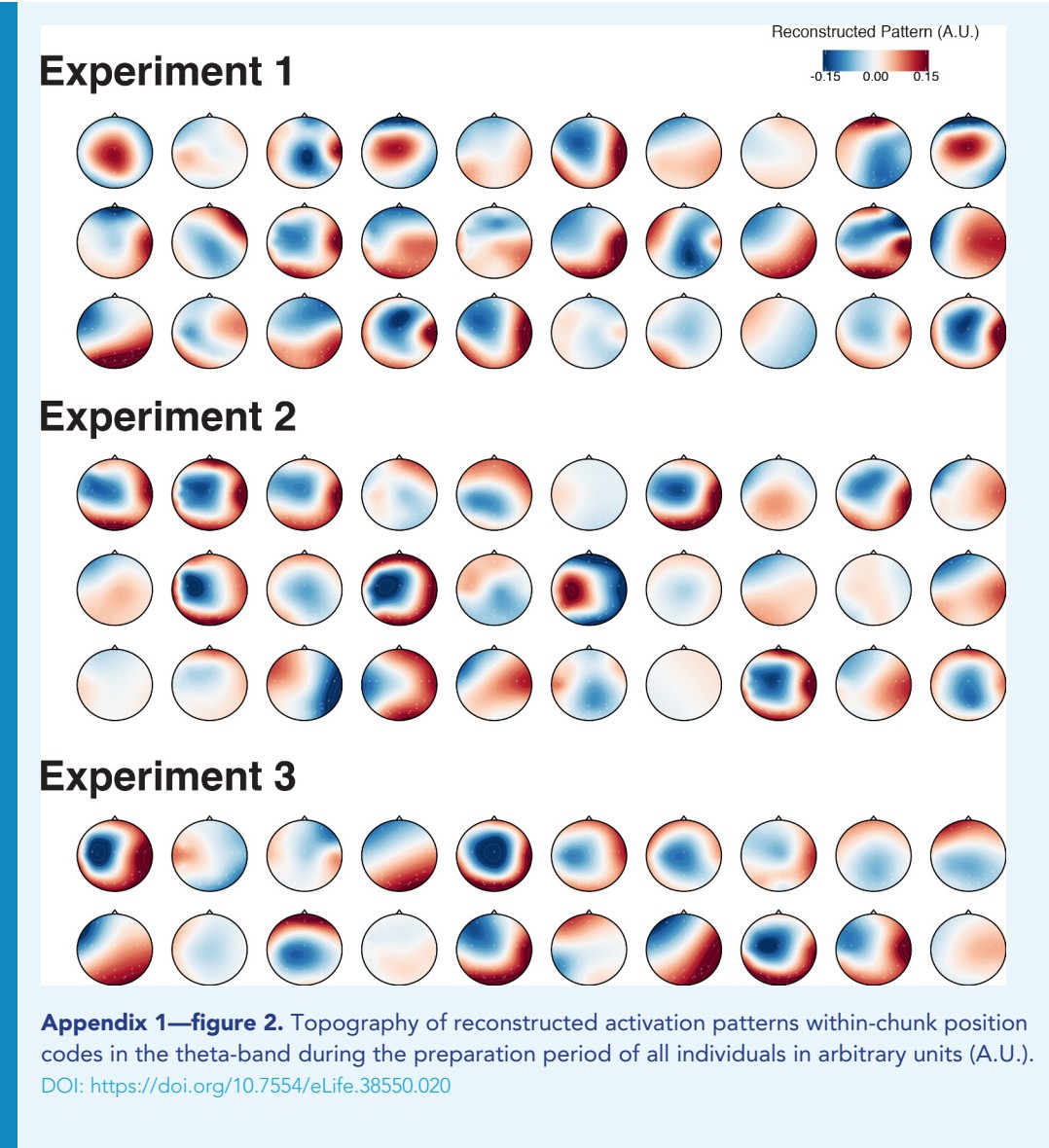

**Appendix 1—figure 2.** Topography of reconstructed activation patterns within-chunk position codes in the theta-band during the preparation period of all individuals in arbitrary units (A.U.).
DOI: https://doi.org/10.7554/eLife.38550.020

