## [Decision Letter]

Thank you for sending your article entitled "Decoding Hierarchical Control of Sequential Behavior in Oscillatory EEG Activity" for peer review at *eLife*. Your article has been evaluated by three peer reviewers, and the evaluation is being overseen by a Reviewing Editor and Sabine Kastner as the Senior Editor.

Given the list of essential revisions, including new experiments, the editors and reviewers invite you to respond within the next two weeks with an action plan and timetable for the completion of the additional work. We plan to share your responses with the reviewers and then issue a binding recommendation.

Summary:

This paper aims to provide evidence for a content-independent, hierarchical representation of action plans in humans. Across three experiments, the study employs multivariate EEG decoding in a paradigm involving judgments about chunked sequences of oriented lines. In addition to being able to decode the identity of individual sequential elements and chunks of elements, the authors find evidence for representations of the ordinal position of both elements and chunks, suggesting that there is an abstract, hierarchical representation of the sequences. The authors find that these representations relate to trial-by-trial reaction time and to individual differences in working memory capacity.

The reviewers were in agreement regarding the merits of this study. The problem of hierarchical control is an important one, and evidence for a central higher-order control signal, like a position code, and its dynamics would be a substantive advance in this domain. Further, the present study has several strengths. For example, the reviewers noted the careful experimental task and design, some control analyses, and the cross-experiment replications.

However, there were also major concerns and limitations noted, some of which relate directly to the main conclusion and contribution. Following discussion, it was decided that a revision to conclusively address these concerns might be difficult, but it is also possible. And so, given the strengths of this study, the decision was to offer you an opportunity to address these concerns in a revision. It should be noted that a revision at *eLife* should fully address the reviewer concerns without need for further revision. Please keep this in mind as you decide whether to resubmit here or start fresh at another journal.

Essential revisions:

1) As noted above, clear evidence for a hierarchical control signal, distinct from the content of individual steps in a sequence would be a very important contribution. However, the reviewers were unconvinced that the results provide evidence for a distinct position code. There was consensus among reviewers that these concerns would need to be addressed decisively for the paper to be acceptable at *eLife*, such as by conducting the proposed control analyses (see reviewer 2's comment below). Specific reviewer comments with regard to this point are provided here.

*Reviewer 1:*

- My major comment is regarding the strong claim that this study provides evidence for a hierarchical control representation, as in a separate, central position code. I think the argument needs further elaboration to be convincing. Botvinick and Plaut (2004) made an interesting insight about higher order representations in their modeling. They showed that the distributed neural representation of a specific sequence element could, nevertheless, share a similarity with other elements of that sequence or with shared sequence elements of other sequences. In this way, the neural representations simultaneously carry information about multiple levels of structure. This is how that model can explain slips and sequence errors even though it uses chaining. This also means that if I were to run pattern classification on the neural representations in their model, I should be able to classify the more abstract sequence level coding based on this shared structure, even as other aspects of the neural representation change with each element. In principle, I could even imagine this emerging at different oscillatory frequencies depending on the scale at which these similarities occurred in a neural ensemble. But, even so, in this study, the evidence for such a frequency dissociation is not very strong. There are different dynamics, but there is also evidence that both theta and alpha carry information about the element and the position similarly following the probe. In the preparation period there is more distinction, but in that phase, there may be contamination by other processes making position 1 mostly distinct from the others. So, it would be helpful to better understand what evidence leads to the strong conclusion that these representations must be encoded separately.

*Reviewer 2:*

- The analyses need to go one step further to truly separate content from position. My understanding of the decoding analyses is that all six of the unique sequences that participants learned are included in both the training and test sets for the classifiers. It seems this could, in principle, allow the classifier to base position judgments on element content, assuming that an element's representation is influenced by the elements that come before and after it. This issue could be avoided by training the classifier on element (or chunk) positions for five of the sequences and testing on a sixth sequence. If this analysis works for Experiment 3, that would be especially convincing. Another variant of this analysis that would be very useful would be to train on element position for the first two chunks of all sequences and test on element position for the third chunk.

- Experiment 3 rules out content-based sequence production that relies on the content of the immediately preceding item, but it does not rule out the possibility that participants are relying on a longer past stimulus history. The Botvinick and Plaut model certainly had the property that it could rely on content from more than one time step in the past. I think this point deserves more discussion. The analyses suggested above (if successful) will demonstrate that abstract position representation is at least PART of the story, but I don't think these experiments can rule out an additional role for content-based sequence production.

*Reviewer 3:*

- In my experience, pattern classification is notorious for pulling out all differences between conditions – meaning every possible confound contributes to the discriminating function. This is a major inferential issue that I don't think is well-appreciated throughout the field. It isn't stated how the authors controlled for confounds like order effects within their task. Having a major temporal confound in the structure makes it impossible to determine what aspects are hierarchical order-specific and which ones are just "later". MVPA could just be picking up on the fact that trial 1 of new chunks have a task-switch type activity that differs from trial 2 and 3. Notably, this is not evidence of hierarchical coding. The RSA data (subsection “Ruling Out Processing-Demand Confounds”) seems to support this more parsimonious and facile explanation.

- The report reads like the alpha and theta findings reflect hierarchical information content in the signals. This is a bold claim – it is very hard to determine the information content of a signal. True evidence for "decoding hierarchical control" would require both sensitivity (which may be reported here) and specificity (in that it does not reflect other processes). I do not think there is enough evidence to suggest the specificity of the information content in these signals reported here, thus I do not think the authors can describe these processes as 'decoding' or that "oscillatory activity in the theta band contains information about sequential position codes".

2) The reviewers noted that the methods were missing crucial details in places, particularly the decoding and RSA procedures. The reviewers found it difficult to understand the task, the instruction to participants, the content of the MVPA analysis, or what exactly was done for several analyses and tests. In discussing these points, there was consensus among the reviewers that a substantial revision of the methods and results overall to enhance clarity and detail is essential in order to properly evaluate the claims of the paper. Though this comment extends to the methods as a whole, the reviewers highlighted some specifics that are listed below.

*Reviewer 2:*

- The representational similarity analysis should have a corresponding section in the Materials and methods with more detail.

*Reviewer 3:*

- The MVPA description isn't clear what the exact contrasts are that are tested, either in the Results or the Materials and methods. The authors claim that ("alpha frequency band allowed sustained decoding of orientations"), but it is never stated how this is achieved. Did the authors train classifiers on 90 degree vs. 45 degree orientations, counterbalanced across all chunk and order sequences?

- How did "elements in the alpha band predicted faster trial-to-trial responses over and above constructs", which sounds correlative, but is tested in a t-test. I have no idea what is being tested here.

- It is never explicitly stated how participants are making the trial-by-trial comparisons – I'm presuming that they learned the sequences beforehand (subsection “Task and Stimuli” seems to detail this – but it comes very late in the description).

- It isn't clear how the process was iterated 30 times when it was a leave-one-block-out procedure. It seems that training would be done on the other 3 blocks and the items in the 4th block would be predicted.

3) It would enhance the paper, and potentially strengthen the claims about separate sources of control signals, if spatial information were considered. Two reviewers felt that even though the present study uses EEG, it underestimates the spatial information that this method can provide.

*Reviewer 1:*

- There is little information on the sources of these effects and how they differ over electrodes. This seems important. Though the spatial resolution of EEG is not its strength, there is some information. It would be helpful to know what the topography of activity is here. In this context, I'll add that the control analyses for the contribution of cognitive control looked at theta. However, the studies cited have typically focused specifically on midfrontal theta. So, the source of theta effects would be helpful to see.

*Reviewer 2:*

- The authors state that the decoding approach provides no information about anatomy, but there are actually a couple of good ways to probe location when using classifiers. One is to analyze the classifier weights that were learned for each electrode. Another is to do the analyses on subsets of the electrodes, as in a searchlight approach. It would certainly add to the paper to display topoplots using one or both of these methods.

4) Beyond the consensus points above, each reviewer raised additional major points that should also be addressed in revision.

*Reviewer 1:*

- The WM capacity results are very interesting, as they suggest an important role for WM particularly in tracking the higher order temporal structure. This being said, is it problematic for the generality of this conclusion that the actual task here is a WM task? Previous studies on hierarchical control of sequences like Schneider and Logan have used simple categorization tasks for the sequence elements that are not as clearly dependent on WM itself. So, in this case, perhaps WM capacity is important if one needs to load up a number of items for a sequence of WM tests. But, it would not be as important if one is sequencing a set of number categorizations. Is there a way to rule this out?

- This study makes an important contribution to the hierarchical control literature, but it does seem a bit dismissive in its review of prior cognitive neuroscience work, particularly from fMRI. It is not fully clear that all these studies, such as Desrochers et al., 2015, are equally confounded by things like stimulus adaptation. Further, Desrochers et al. used TMS to show that the contribution of the frontal pole observed with fMRI was not epiphenomenal. So, it might enhance the impact of the present work to try to engage with, rather than dismiss, the prior results on sequential control, particularly as the spatial resolution of fMRI complements the limitations of the EEG results reported here.

- In Experiment 1, it looks like the first chunk has the longest RT. Did this contribution to the classification of chunk identity? Is there a similar confusion matrix for chunk identity?

*Reviewer 2:*

- In Experiment 3, were chunk identity and position (level 2 codes) analyzed? Unless I missed a reason that these were left out, the results of these analyses should be reported.

- The analysis presented in the Discussion section about forward vs. backward predictability seems important. It deserves more explanation and a treatment in the Results section.

- There should be some justification for the use of a one-tailed cluster significance threshold (cluster formation threshold not as important).

---

## [Author Response]

[Editors' note: the authors’ plan for revisions was approved and the authors made a formal revised submission.]

Essential revisions:1) As noted above, clear evidence for a hierarchical control signal, distinct from the content of individual steps in a sequence would be a very important contribution. However, the reviewers were unconvinced that the results provide evidence for a distinct position code. There was consensus among reviewers that these concerns would need to be addressed decisively for the paper to be acceptable at eLife, such as by conducting the proposed control analyses (see reviewer 2's comment below). Specific reviewer comments with regard to this point are provided here.Reviewer 1:My major comment is regarding the strong claim that this study provides evidence for a hierarchical control representation, as in a separate, central position code. I think the argument needs further elaboration to be convincing. Botvinick and Plaut (2004) made an interesting insight about higher order representations in their modeling. They showed that the distributed neural representation of a specific sequence element could, nevertheless, share a similarity with other elements of that sequence or with shared sequence elements of other sequences. In this way, the neural representations simultaneously carry information about multiple levels of structure. This is how that model can explain slips and sequence errors even though it uses chaining. This also means that if I were to run pattern classification on the neural representations in their model, I should be able to classify the more abstract sequence level coding based on this shared structure, even as other aspects of the neural representation change with each element. In principle, I could even imagine this emerging at different oscillatory frequencies depending on the scale at which these similarities occurred in a neural ensemble. But, even so, in this study, the evidence for such a frequency dissociation is not very strong. There are different dynamics, but there is also evidence that both theta and alpha carry information about the element and the position similarly following the probe. In the preparation period there is more distinction, but in that phase, there may be contamination by other processes making position 1 mostly distinct from the others. So, it would be helpful to better understand what evidence leads to the strong conclusion that these representations must be encoded separately.Reviewer 2:- The analyses need to go one step further to truly separate content from position. My understanding of the decoding analyses is that all six of the unique sequences that participants learned are included in both the training and test sets for the classifiers. It seems this could, in principle, allow the classifier to base position judgments on element content, assuming that an element's representation is influenced by the elements that come before and after it. This issue could be avoided by training the classifier on element (or chunk) positions for five of the sequences and testing on a sixth sequence. If this analysis works for Experiment 3, that would be especially convincing. Another variant of this analysis that would be very useful would be to train on element position for the first two chunks of all sequences and test on element position for the third chunk.- Experiment 3 rules out content-based sequence production that relies on the content of the immediately preceding item, but it does not rule out the possibility that participants are relying on a longer past stimulus history. The Botvinick and Plaut model certainly had the property that it could rely on content from more than one time step in the past. I think this point deserves more discussion. The analyses suggested above (if successful) will demonstrate that abstract position representation is at least PART of the story, but I don't think these experiments can rule out an additional role for content-based sequence production.Reviewer 3:- In my experience, pattern classification is notorious for pulling out all differences between conditions – meaning every possible confound contributes to the discriminating function. This is a major inferential issue that I don't think is well-appreciated throughout the field. It isn't stated how the authors controlled for confounds like order effects within their task. Having a major temporal confound in the structure makes it impossible to determine what aspects are hierarchical order-specific and which ones are just "later". MVPA could just be picking up on the fact that trial 1 of new chunks have a task-switch type activity that differs from trial 2 and 3. Notably, this is not evidence of hierarchical coding. The RSA data (subsection “Ruling Out Processing-Demand Confounds”) seems to support this more parsimonious and facile explanation.- The report reads like the alpha and theta findings reflect hierarchical information content in the signals. This is a bold claim – it is very hard to determine the information content of a signal. True evidence for "decoding hierarchical control" would require both sensitivity (which may be reported here) and specificity (in that it does not reflect other processes). I do not think there is enough evidence to suggest the specificity of the information content in these signals reported here, thus I do not think the authors can describe these processes as 'decoding' or that "oscillatory activity in the theta band contains information about sequential position codes".

Combined response: We are grateful for these constructive and theoretically important points that get to the heart of what we are trying to achieve. Allowing us to provide additional evidence and to clarify our conclusions (including their limitations) has considerably strengthened our manuscript. Specifically, we took the following steps:

1) We provide additional evidence that element and position codes are indeed independent of each other. Specifically, in all three experiments, we now show that position decoding generalizes across instances of different elements, chunk positions, and in Experiment 3 also chunk identity (which could not be adequately tested in Experiments 1 and 2; see Tables 1 and 3).

2) The sequences used in Experiment 3 allowed us to apply RSA to directly test for conjunctions between positions and elements. We found very little evidence for such representations (Figure 11, subsection “Independence of Position and Element Codes?”). Both 1) and 2) directly respond to the queries by Reviewer 2 (in particular the Generalization analysis), but they are also relevant for Reviewer 1’s questions related to the Botvinick and Plaut model (which predicts the emergence of integrated representations), and Reviewer 3’s question about the specificity of the decoded information.

3) An important prediction of the Botvinick and Plaut model is the emergence of integrated representations of successive elements, which according to the model should serve as primary predictors of future elements. Again, we used RSA to directly test this prediction in Experiment 3, finding no evidence for such representations (Figure 12, subsection “Integrated Representations of Successive Elements?”).

4) We emphasize both in the Introduction and in the Discussion that our paradigm was geared towards revealing independent control codes (e.g., by using new sequences in every block and a paradigm with strong, previous behavioral evidence suggesting position codes, Mayr, 2009). As we now clearly state, our results provide positive evidence that independent control codes handle sequencing demands in certain situations, but do not rule out recurrent-network-type representations in other circumstances (subsections “Overview” and “Working Memory Constraints”).

5) As suggested by reviewer 2, we include the Shapiro et al. chaining analyses in the Results section (subsection “Evidence for Associative Chaining?”).

6) The confound issue raised by reviewer 3 was indeed a very important concern that we tried to respond to both in our procedures and our analyses.

We hope that we now better communicate how we tried to rule out obvious confounds through our procedures. Specifically, all position-element combinations occurred equally often within each sequence (Experiment 1 and 2) or across sequences (Experiment 3), and in Experiment 1 and 2 also all chunks occurred in each sequence. Further, the “cycling” procedure ensured that there was no confound between specific positions and the temporal distance to the beginning or end of the test phase (one of the major confounds mentioned by Kalm and Norris).

The one remaining issue is indeed the chunk-switch/retrieval confound that potentially makes position 1 different from positions 2 and 3. We worked hard to eliminate/reduce this confound by promoting early retrieval (Experiment 1), using self-paced retrieval prior to the preparation phase (Experiment 2), greater pre-training (Experiment 3), and by separately analyzing the preparation and the probe period. At least for the probe period, our decoding results were highly robust across the different experiments, despite the variation in processing demands. In addition, the goal of the RSA analyses was exactly to determine to what degree position 1 stands out, potentially indicating a switch/retrieval confound. These analyses indicate that during the preparation period in Experiments 1 and 2, position 1 is indeed special (Figure 4). However, there was no evidence for a unique-position-1 effect for the probe period in Experiments 1 and 2, and in Experiment 3 for neither the preparation or the probe period (Figure 10).

We provide arguments why the preparation-period patterns in Experiments 1 and 2 do not necessarily reflect retrieval demands, but may very well reflect a representation that differentiates between the first and subsequent positions (subsections “Within-Chunk Positions and Processing-Demand Confounds”, “Individual Differences and WM Capacity”). For example, a retrieval confound explanation would suggest that for people with low WM capacity (who have the largest retrieval difficulties at chunk boundaries), such a confound should be particularly strong and therefore should show more robust position decoding than high WM individuals. We found no such evidence. Importantly, even if one does not buy our indirect arguments regarding the preparation period, the probe-period effects cannot be easily explained in terms of position confounds. Thus, at least for this period, we feel confident that our decoding results reflect position information.

2) The reviewers noted that the methods were missing crucial details in places, particularly the decoding and RSA procedures. The reviewers found it difficult to understand the task, the instruction to participants, the content of the MVPA analysis, or what exactly was done for several analyses and tests. In discussing these points, there was consensus among the reviewers that a substantial revision of the methods and results overall to enhance clarity and detail is essential in order to properly evaluate the claims of the paper. Though this comment extends to the methods as a whole, the reviewers highlighted some specifics that are listed below.

In response to this general point, we rewrote the Materials and methods section (adding among other aspects, specific sections for all major analysis components), we tried to provide a better “high-level” description of our procedure at the beginning of the Results section, we added information to the figure legends (in particular Figure 1), and we tried to provide additional high-level information about our analyses in the Results section.

Reviewer 2:- The representational similarity analysis should have a corresponding section in the Materials and methods with more detail.

We added such a section (“Representational Similarity Analysis”).

Reviewer 3:- The MVPA description isn't clear what the exact contrasts are that are tested, either in the Results or the Materials and methods. The authors claim that ("alpha frequency band allowed sustained decoding of orientations"), but it is never stated how this is achieved. Did the authors train classifiers on 90 degree vs. 45 degree orientations, counterbalanced across all chunk and order sequences?

We present the analysis in greater detail (subsection “Frequency-by-Time Decoding Analysis”).

- How did "elements in the alpha band predicted faster trial-to-trial responses over and above constructs", which sounds correlative, but is tested in a t-test. I have no idea what is being tested here.

We describe our multi-level modeling analyses in a separate section (subsection “Multi-Level Modeling”).

- It is never explicitly stated how participants are making the trial-by-trial comparisons – I'm presuming that they learned the sequences beforehand (subsection “Task and Stimuli” seems to detail this – but it comes very late in the description).

We added information on our procedures at the beginning of the Results section, to the legend of Figure 1, and the Materials and methods section.

- It isn't clear how the process was iterated 30 times when it was a leave-one-block-out procedure. It seems that training would be done on the other 3 blocks and the items in the 4th block would be predicted.

These details are now presented in the subsection “Frequency-by-Time Decoding Analysis”.

3) It would enhance the paper, and potentially strengthen the claims about separate sources of control signals, if spatial information were considered. Two reviewers felt that even though the present study uses EEG, it underestimates the spatial information that this method can provide.Reviewer 1:- There is little information on the sources of these effects and how they differ over electrodes. This seems important. Though the spatial resolution of EEG is not its strength, there is some information. It would be helpful to know what the topography of activity is here. In this context, I'll add that the control analyses for the contribution of cognitive control looked at theta. However, the studies cited have typically focused specifically on midfrontal theta. So, the source of theta effects would be helpful to see.Reviewer 2:- The authors state that the decoding approach provides no information about anatomy, but there are actually a couple of good ways to probe location when using classifiers. One is to analyze the classifier weights that were learned for each electrode. Another is to do the analyses on subsets of the electrodes, as in a searchlight approach. It would certainly add to the paper to display topoplots using one or both of these methods.

Response to reviewers 1 and 2: We now present topographic information in the supplementary material based on the procedure 1 suggested by reviewer 2. These plots are not hugely informative and essentially confirm that much of the decoding-relevant information is rather idiosyncratic--in particular in the probe period. Given that we are trying to decode relatively abstract features, we believe that his is not very surprising.

4) Beyond the consensus points above, each reviewer raised additional major points that should also be addressed in revision.Reviewer 1:- The WM capacity results are very interesting, as they suggest an important role for WM particularly in tracking the higher order temporal structure. This being said, is it problematic for the generality of this conclusion that the actual task here is a WM task? Previous studies on hierarchical control of sequences like Schneider and Logan have used simple categorization tasks for the sequence elements that are not as clearly dependent on WM itself. So, in this case, perhaps WM capacity is important if one needs to load up a number of items for a sequence of WM tests. But, it would not be as important if one is sequencing a set of number categorizations. Is there a way to rule this out?

While we are not sure we agree with the premise, we are grateful for this query, as it prompted us to consider the generality of our results (subsection “Working Memory Constraints”). We had chosen orientations because there is prior evidence that these can be decoded via EEG, whereas no such evidence is available for “tasks” as basic elements. The reviewer’s premise seems to assume that it matters for information load or type of representation whether subjects need to reproduce a short sequence of orientations (that require on each trial require a simple match/no-match response) or a short series of types of categorizations (e.g., odd/even, smaller/larger 5) as in the standard task-span procedure. We believe that it is likely that the latter actually imposes greater WM demands as it also requires a representation of arbitrary S-R rules for each task. In fact, average RTs in our task are considerably lower than in studies using the task-span procedure, suggesting that if anything our paradigm imposes a smaller load than the regular procedure. We now discuss these aspects and argue that the working-memory demands of the primary elements (and also the chunk size) may determine to what degree a neat dissociation between working memory effects on within-chunk versus chunk-level representations can be found.

- This study makes an important contribution to the hierarchical control literature, but it does seem a bit dismissive in its review of prior cognitive neuroscience work, particularly from fMRI. It is not fully clear that all these studies, such as Desrochers et al., 2015, are equally confounded by things like stimulus adaptation. Further, Desrochers et al. used TMS to show that the contribution of the frontal pole observed with fMRI was not epiphenomenal. So, it might enhance the impact of the present work to try to engage with, rather than dismiss, the prior results on sequential control, particularly as the spatial resolution of fMRI complements the limitations of the EEG results reported here.

Our intention was not to be dismissive of earlier fMRI work; we only wanted to emphasize the unique challenges when looking at serial-order in an element-by-element manner using fMRI. We agree that the Desrochers et al. study gets around the adaptation confound because it also uses a “cycling-sequence” paradigm and clearly the TMS findings provide strong corroboration of the imaging results. One might note though that the imaging task required an inter-trial interval of 16 seconds, which might raise some questions about comparability to more standard sequencing situations. Overall, we now strived for a more nuanced presentation of earlier fMRI work.

- In Experiment 1, it looks like the first chunk has the longest RT. Did this contribution to the classification of chunk identity? Is there a similar confusion matrix for chunk identity?

Note: any of the three chunks occurred equally often at any position within the sequence. Thus, chunk position could not have contributed to decoding of identity. We have tried to further emphasize the mutual independence of the four different features (element identity and order, chunk identity and order, see for example subsection “Overview”). For chunk order however, it is in fact possible that the decoding accuracy might be driven by retrieval confounds and therefore in the revision we are even more cautious now in drawing strong conclusions from these results (subsection “Qualifications”).

*Reviewer 2:*

- In Experiment 3, were chunk identity and position (level 2 codes) analyzed? Unless I missed a reason that these were left out, the results of these analyses should be reported.

A key benefit of Experiment 1 and 2 was that each expression of each of the four key features (element identity and order, chunk identity and order) was present in each sequence. This has great advantages for decoding as it eliminates the possibility that block context can affect decoding accuracy. As Experiment 3 was geared towards analyzing within-chunk positions and elements without the “chaining confound” present in Experiments 1 and 2 we had to change the number of possible chunk identities (6) and chunk positions (2). In particular for chunk identity this meant that we had to give up on having all chunks present within each sequence. We therefore felt that we did not have a good basis for decoding this feature. In addition, we had not assessed WM capacity in this experiment. We now better explain the reason why we focus only on positions and elements in this experiment (subsection “Experiment 3”).

- The analysis presented in the Discussion section about forward vs. backward predictability seems important. It deserves more explanation and a treatment in the Results section.

We moved this analysis into the Results subsection “Evidence for Associative Chaining”.

- There should be some justification for the use of a one-tailed cluster significance threshold (cluster formation threshold not as important).

We believe that in this situation, a one-tailed threshold is appropriate as we are only interested in identifying clusters significantly above chance. We now state so in the Materials and methods section (subsection “Cluster-Based Permutation Test”).